# Coupling of numerical groundwater-ocean models to improve understanding of the coastal zone

Jiangyue Jin[1,2,3], Manuel Espino[1,3], Daniel Fernández-Garcia [1,2], Albert Folch[1,2]

[1]Department of Civil and Environmental Engineering (DECA), Universitat Politècnica de Catalunya (UPC), Barcelona 08034, Spain

[2]Associated Unit: Hydrogeology Group (UPC-CSIC), Spain

[3]Laboratori d'Enginyeria Marítima (LIM)

*Correspondence to*: Jiangyue Jin (Jiangyue.Jin@upc.edu)

**Abstract.** Coastal zones are increasingly acknowledged as dynamic yet fragile components of global ecosystems amidst escalating anthropogenic activities and complex land-ocean interactions. Understanding the interactions between groundwater and the ocean is crucial for managing submarine groundwater discharge (SGD) and seawater intrusion (SWI), vital for coastal ecosystem preservation and water resource management. This research proposes an integrated modeling approach that couple groundwater flow and physical oceanographic models to accurately simulate coastal-ocean groundwater interactions.

In this work, a TELEMAC-3D based three-dimensional hydrodynamic model was initially developed to capture marine conditions with variable salinity and temperature. A MODFLOW6 groundwater model was subsequently constructed. The models were efficiently coupled using Flopy and Telapy, enabling precise co-simulation of hydrodynamic and groundwater systems. Validation of the coupled model against empirical data confirmed its high fidelity, with errors within acceptable ranges.

This coupled model employs dynamic boundary conditions, overcoming the limitations of traditional coastal groundwater models that assume constant salinity. This enhancement significantly improves the accuracy and practicality of simulating SGD processes in the coastal ocean. The bidirectional feedback mechanism within the coupled model strengthens the analysis of interactions between the ocean and groundwater systems. It accounts for variations in the seawater boundary under tidal influence and the reciprocal impact of groundwater dynamics on the hydrodynamic conditions of nearshore waters. This holistic enhancement bolsters the model's hydrological simulation capabilities, providing a more comprehensive depiction of the intricate water-salt exchange mechanisms in coastal systems.

**Keywords** Coastal zone modeling, SGD, SWI, Coupled model, Telemac, Telapy, Modflow6, Flopy, Groundwater-ocean interaction

## 1 Introduction

The coastal zone, a critical ecological interface where land and sea intersect, carries a unique ecological environment and important functions in the global ecosystem(Turner et al., 1996). Its dynamic balance is increasingly affected by human activities and the intensification of land-sea interactions, making it one of the most vibrant and sensitive parts of the Earth(Ramesh et al., 2015). Faced with the dual pressures of environmental change and human activities, it is particularly crucial to deeply understand the interactions among various parts of the Earth's water cycle system, especially in the coastal zone where the ocean and land meet. Among them, the interaction between the ocean and the terrestrial groundwater system, especially seawater intrusion (SWI) (Kim et al., 2015) and submarine groundwater discharge (SGD)(Lin et al., 2024), has profound impacts on the hydrological cycle, water resource quality(Santos et al., 2021), ecosystem health, and global material cycle in the coastal zone(Cao et al., 2021). Submarine Groundwater Discharge (SGD) comprises two components: Freshwater SGD (FSGD) from inland aquifers and Recirculated SGD (Wilson) driven by tidal pumping or wave action(Burnett et al., 2003).

As the phenomenon of seawater intrusion caused by over-extraction of groundwater intensifies, marine ecosystems are facing serious threats, manifested as the decline of ecosystems such as bays, estuaries, and coastal wetlands. At the same time, groundwater pollution (Perumal et al., 2024) is an urgent problem to be solved, which not only affects water quality, but may also introduce pollutants into the ocean through the SGD pathway, further deteriorating seawater quality and causing long-term damage to the ecosystem(Moore & Joye, 2021). In addition, eutrophication of water bodies (Dong et al., 2024) and hypoxic events (Wang et al., 2022) have also become severe challenges facing the current coastal environment, exacerbating the pressure on aquatic ecosystems.

In the face of these challenges, researcher are concurrently focusing on the interactions between groundwater systems(Martínez-Pérez et al., 2022) and marine ecosystems(Ramatlapeng et al., 2021) to deeply analyze the intrinsic connection between groundwater dynamic changes and marine ecological problems (Fang et al., 2021a). Developing models that accurately represent the dynamic interactions between groundwater and ocean systems is essential for a deeper understanding and better management of these interconnected environments.

Although traditional coastal groundwater studies have considered boundary conditions such as tidal changes and sea level variations when assessing SWI(Yu et al., 2019). Most research still treats the ocean as a static boundary, overlooking the dynamic impact of ocean dynamics on groundwater migration rules(Nguyen et al., 2020). Similarly, oceanographers highly value factors such as waves, tides, and currents when exploring the interface processes between surface water and the ocean. However, they often overlook the potential contribution of groundwater as an important terrestrial water source to the chemical composition, thermodynamic state, and ecosystem functions of the ocean. When constructing ocean models, the influence of groundwater is typically not considered(Arévalo-Martínez et al., 2023). Therefore, developing comprehensive models that can accurately depict these complex dynamic processes is crucial for effective management and quantification of internal processes both in coastal waters and in coastal aquifers.

Recent interdisciplinary research has clearly pointed out that relying solely on independent surface water models or groundwater models is insufficient to fully reveal the essential characteristics of complex hydrological processes in coastal zones(Arévalo-Martínez et al., 2023). Especially the bidirectional coupling between groundwater and the ocean(Dassargues et al., 1996), including the impact of groundwater on the marine environment and the feedback effect of ocean dynamics on the dynamics of the groundwater system, is not fully expressed in traditional separate models(Lewandowski et al., 2020). Therefore, the development of coupled models that can simultaneously simulate and integrate the interactions between groundwater and the ocean has become an urgent task in academic research.

In the practice of constructing ocean-groundwater coupled models, researchers have encountered a series of significant technical challenges (Haque et al., 2021). First, how to solve the problem of system scale and dynamic differences is a key difficulty in integration(Carabin & Dassargues, 1999). When building ocean models, groundwater is usually not considered, rapid hydrological phenomena such as tides need to be simulated with high accuracy, which usually requires a fine spatial grid resolution; in contrast, groundwater models focus on groundwater flow movement at larger scales, paying attention to relatively slow hydrological cycles, resulting in prominent problems of mismatched grid size and time scale. At the same time, achieving

synchronous operation of coupled models on different time frames is also a daunting task(Yang et al., 2013a). Ocean models often track rapidly changing ocean dynamics on an hourly scale, while the calculation period of groundwater models may be in days or weeks. To realistically simulate the interaction between the two in the actual environment, it is urgent to find effective means to coordinate the simulation consistency of the two models at different time and space scales, to overcome the aforementioned integration problems and to ensure that the coupled models can accurately describe complex hydrological processes, salinity distribution, pollutant transport, and nutrient cycling in coastal areas(Liu et al., 2023). Therefore, beyond resolving spatial and temporal inconsistencies, it is also essential to refine model structures to better simulate dynamic boundary conditions and interface processes.

Against this backdrop, early coupled models, such as Yuan et al(2011), attempts to integrate surface water and groundwater interactions in coastal wetlands. However, these models primarily focused on regional-scale processes and relied on simplified boundary conditions. While they provided a solid foundation, they lacked the capability to dynamically adjust salinity and pressure at the land-sea interface under tidal influence. Later frameworks, such as HydroGeoSphere(Brunner & Simmons, 2012), improved multi-scale hydrological simulations but faced challenges in resolving steep salinity gradients and bidirectional tidal-groundwater feedback due to computational and parameterization constraints. Similarly, coupling surface water models with SUTRA (Voss & Provost, 2002) enabled variable-density flow simulations but often treated the ocean as a static boundary, overlooking the dynamic interplay between seawater intrusion (SWI) and SGD. Despite these advancements, a major challenge remains in accurately representing the dynamic exchange processes at the groundwater-seawater interface.

Recent advancements in coupled modeling have sought to overcome these limitations. Our TELEMAC-MODFLOW framework introduces three key innovations to address these gaps. First, it dynamically updates boundary salinity and temperature based on real-time flux direction, allowing the coastal interface to transition between submarine groundwater discharge (SGD) and seawater intrusion (SWI), thereby resolving transient mixing zones. Second, it incorporates bidirectional feedback between tidal fluctuations and groundwater flow, ensuring that tidal pumping effects on SGD rates and nearshore stratification are accurately captured. Finally, by coupling TELEMAC-3D's non-hydrostatic solver with MODFLOW 6's density-dependent flow module, the model achieves high-resolution simulation of aquifer-ocean interactions, crucial for quantifying recirculated SGD (RSGD).

At the interface between groundwater and seawater, due to the physical property differences of the media (seawater flows in an open marine environment, while groundwater flows in porous media) and the momentum differences between seawater and groundwater, the speed of seawater entering groundwater may significantly decrease, while the impact of groundwater entering seawater on the speed of seawater may be relatively small(Slomp & Van Cappellen, 2004a). Therefore, when simulating the interaction process between groundwater and seawater, we may not need to consider the speed changes at the interface in detail. However, this does not mean that speed has no impact on the interaction process between groundwater and seawater. For instance, changes in speed may affect the transport and diffusion of substances, thereby affecting the material exchange between groundwater and seawater(Michael et al., 2005).

The aim of this study is to propose a coupling method between an ocean hydrodynamic model and a groundwater flow model while quantitatively assessing the interaction between groundwater (GM) and oceanic dynamics (OM). In this research, we constructed a three-dimensional hydrodynamic model based on TELEMAC-3D (Hervouet, 2007) to simulate coastal OM. Simultaneously, a GM using MODFLOW6(Hughes et al., 2017) was developed. To facilitate the coupling of these models, we have employed the Telapy library(Audouin et al., 2017), a Python package designed to interface with Telemac, and Flopy(Bakker et al., 2016), a set of Python modules that provide a powerful means of pre- and post-processing MODFLOW models. These tools have been instrumental in enabling a loosely coupled approach, where the two models exchange information and synchronize their simulations in an iterative process.

To validated the constructed model, different laboratory experiment published in the scientific literature studies has been simulated confirm the accuracy and reliability of the coupled ocean and groundwater model. By validating the coupled model at the laboratory scale and further building on the coupled model under tidal conditions, we aim to ensure its robustness and applicability for broader, real-world scenarios, thus contributing to the advancement of coastal aquifer research and management.

1  The current model focuses on groundwater-ocean interactions (e.g., tidal effects and SGD) and does not resolve porewater flow

2  processes such as wave-driven shear flow or sediment-water interface dynamics (Huettel et al., 2014; Taniguchi et al., 2019).

## 2 Coupling of numerical groundwater-ocean models

### 2.1. Groundwater Model

#### 2.1.1. MODFLOW 6

MODFLOW 6 is an object-oriented program that facilitates the integration of multiple models within a single simulation framework(Hughes et al., 2017). This sixth core version by the USGS supports independent operation and information exchange between models, enabling various interactions. The program is based on the principle of integrated finite differences to calculate hydraulic head within a central grid, which is essential for simulating complex groundwater flow scenarios, including saltwater intrusion.

In the context of groundwater flow, Darcy's Law is fundamental in describing fluid movement. For conditions of variable density, the hydraulic head $(\nabla h)$ form of Darcy's Law is expressed as:

$$q = -K_0 \left( \nabla h + \frac{\rho - \rho_0}{\rho_0} g \nabla z \right) \quad (1)$$

Here, $q$ represents the specific discharge vector, $K_0$ is the hydraulic conductivity, $\rho$ is the local fluid density, $\rho_0$ is the reference density, $g$ is the acceleration due to gravity, and $\nabla z$ is the gradient of elevation.

MODFLOW 6 allows for the simulation of groundwater flow that accounts for variations in water density without the need to convert between freshwater head and hydraulic head. Correction term is added directly to the calculations based on constant-density flow to reflect the effects of density changes(Langevin et al., 2020a). This enhancement simplifies the simulation process and improves the flexibility and accuracy in handling complex groundwater flow issues.

### 2.1.2 Description of MODFLOW API

The MODFLOW API(Hughes et al., 2022) has been instrumental in refining the boundary conditions and salinity values within our groundwater model. By leveraging the API's Basic Model Interface(BMI) capabilities, we efficiently adjusted the General Head Boundary (GHB) package to simulate dynamic coastal interactions without altering the source code. This flexibility allowed for precise control over boundary head specifications, essential for capturing tidal influences and sea-level adjustments.

Moreover, the seamless integration with the Flopy library(Bakker et al., 2016) facilitated the coupling of the groundwater model with an ocean model. Through this integration, the models exchanged critical data, such as salinity gradients and head levels, enabling a holistic simulation of the groundwater-ocean system. This coupling was pivotal in understanding the complex interplay between groundwater and ocean dynamics, particularly in the context of saltwater intrusion.

The combined use of MODFLOW API and Flopy streamlined the modeling process, providing a robust framework for analyzing and managing coastal aquifer systems under varying salinity conditions.

### 2.2. Ocean Model

#### 2.2.1. Description of TELEMAC-3D Model

1  The TELEMAC-3D model, part of the TELEMAC software suite(Hervouet, 2007), is a sophisticated Computational Fluid
2  Dynamics (CFD) tool designed for simulating diverse aquatic environments. The model accurately captures free-surface
3  dynamics through the solution of the non-hydrostatic Navier-Stokes equations, which are expressed as:

$$\begin{cases} \rho\left(\dfrac{\partial u}{\partial t} + u \cdot \nabla u\right) = -\nabla p + \nabla\tau - \nabla hp + \nabla h \cdot \tau h + \rho f_c(k \times u) \\ \qquad \rho\left(\dfrac{\partial w}{\partial t} + w \cdot \nabla w\right) = -\dfrac{\partial z}{\partial p} + \dfrac{\partial \tau_z}{\partial z} - \rho g \end{cases} \qquad (2)$$

In this equation, Where: u is the horizontal velocity vector, w is the vertical velocity, $\nabla h$ is the horizontal gradient operator, $\tau_h$
is the horizontal stress tensor parameterized using the horizontal eddy viscosity coefficient $\tau_z$ is the vertical turbulent stress
parameterized using the vertical eddy viscosity coefficient, fc is the Coriolis parameter, g is the gravitational acceleration, k is
the vertical unit vector.

To incorporate temperature and salinity into the TELEMAC-3D model, the fluid density $\rho$ is modified as:

$\rho = \rho_0\left(1 - \alpha_T(T - T_0) + \beta_S(S - S_0)\right)$  (3)

Where $\alpha_T$ is the thermal expansion coefficient, $\beta_S$ is the haline contraction coefficient, $T$ is temperature and $S$ is salinity.

The buoyancy force $f_b$ in the Navier-Stokes equation, which governs the fluid dynamics, is adjusted to include temperature
and salinity effects:

$f_b = \rho_0 g\left(\alpha_T(T - T_0) - \beta_S(S - S_0)\right)$  (4)

Model configuration is streamlined through the use of keywords in the ".cas" control file, which defines topography, boundary
conditions, and other essential parameters. The choice of time step is critical for the model's temporal resolution and overall
simulation accuracy, ensuring a faithful representation of the coupled groundwater-ocean system.

**2.2.2. Description of Telapy**

TelApy is a Python wrapper for the TELEMAC API that offers precise control over simulations (Audouin et al., 2017). It
enables users to pause simulations, access specific variables, and modify their values using a Fortran structure called
"instantiation." By adjusting TELEMAC's main subroutines, TelApy allows for step-by-step execution of hydraulic cases.
Python's versatility, portability, and extensive libraries make it an ideal tool for driving TELEMAC-MASCARET SYSTEM
APIs, enhancing simulation control and efficiency.

The development of TelApy addresses the complexity and accessibility challenges users faced with the Fortran API. With
TelApy, users can leverage Python's flexibility and expandability to simplify the simulation setup and execution process. The
rich library support in Python, such as NumPy and SciPy, facilitates complex data processing and analysis, making it a
powerful and intuitive tool for researchers and engineers in hydrological modeling and marine engineering.

**2.3. Coupling Modflow and Telemac**

**2.3.1. Method Overview**

In our research, we propose a bidirectional dynamic coupling method that can achieve a close integration of the ocean model and the groundwater model. Specifically, with the help of the respective Python interfaces of Telemac and MODFLOW (Telapy based on Telemac, Flopy based on MODFLOW), we have designed a coordination system that allows the two models to alternate step-by-step during the simulation process and share necessary boundary condition information in real-time. This includes parameters such as hydraulic head, temperature and salinity. Although the current implementation does not include the simulation of solute pollutants, the coupling framework is designed to support the inclusion of these variables in future studies. This coupling method aims to enhance the accuracy and reliability of simulation results by accurately reflecting the interaction of the ocean-groundwater system.

In simulations of groundwater and seawater interaction, the traditional Henry problem assumes a constant salinity at the coastal boundary (Langevin et al., 2020b). However, we opted to refine this setup by designing the coastal boundary as a combination of a discharge area and a region maintaining seawater salinity. Yang introduced an adaptive salt mass flux boundary, which adjusts salinity based on the flow direction(Yang et al., 2013b).Consequently, at each time step, the boundary salinity is determined by the flow direction at the groundwater model's boundary conditions.. The coupling of the MODFLOW6 groundwater model and the TELEMAC-3D ocean model hinges on a bidirectional exchange of hydraulic head, salinity, and temperature at the land-sea interface. This dynamic interaction is governed by the direction of groundwater flux (Q), determined at each time step. When Q>0 (groundwater discharge into the ocean), TELEMAC adopts MODFLOW's salinity (Sg) and temperature (Tg) as boundary conditions. Conversely, during seawater intrusion (Q<0), MODFLOW updates its coastal boundaries using TELEMAC's outputs (So, To).

In MODFLOW6, the Groundwater Transport (GWT) module simulates solute and heat transport via the advection-dispersion equation (ADE) (Langevin et al., 2020):

$$\frac{\partial v}{\partial t}(\theta C) = \nabla \cdot (\theta D \nabla C) - \nabla \cdot (qC) + Qs \quad (5)$$

where C represents salinity or temperature, $\theta$ is porosity, D is the dispersion tensor, q is Darcy velocity, and Qs denotes source/sink terms. The dispersion tensor accounts for longitudinal ($\alpha L$) and transverse ($\alpha T$) dispersity, critical for resolving anisotropic mixing in coastal aquifers.

TELEMAC-3D resolves marine dynamics using the 3D Navier-Stokes equations coupled with turbulent diffusion(Hervouet, 2007):$\frac{\partial C}{\partial t} + u \cdot \nabla C = \nabla \cdot (v t \nabla C) + S$  (6)

$$\frac{\partial C}{\partial t} + u \cdot \nabla C = \nabla \cdot (v t \nabla C) + S \quad (6)$$

where u is the velocity field, $v_t$ is turbulent diffusivity, and S represents external sources. Horizontal mixing is driven by grid-scale advection and subgrid turbulence, while vertical mixing incorporates buoyancy effects through a k-$\epsilon$ turbulence closure model. Due to the significant scale disparity between groundwater Darcy velocities (ranging from $10^{-6}$ to $10^{-4}$ m/s) and oceanic flow velocities (on the order of $10^{-2}$ to 1 m/s), direct simulation of mass flux exchange across the interface is not performed. Instead, salinity and temperature profiles are implemented as Dirichlet boundary conditions. Subsequently, mixing processes are simulated through turbulent diffusion within TELEMAC and hydrodynamic dispersion within MODFLOW. For example, during groundwater discharge events, the salinity of the discharging groundwater (SGW) is imposed at TELEMAC's seabed boundary, and its subsequent dilution by turbulent mixing emulates natural dispersion processes, as detailed by Slomp & Van Cappellen(2004b).

Our approach includes different key steps (fig 1):

1)At the start of computation, the model first obtains the state of the marine and groundwater models at their interface, including
variables such as water level, salinity, and temperature.

2). Using the Telapy and Flopy interfaces, the temperature and salinity information of the marine model is transferred to the
corresponding points of the groundwater model, and the information of the groundwater model is transferred to the corresponding
points of the marine model. Data synchronization was achieved by directly transferring data between grid points that were
spatially coincident in both model domains, with iterative boundary condition updates governed by interface flow direction.

3). A single time step calculation is performed for the marine and groundwater models. Throughout the process, we pay special
attention to the exchanged variables, including water level, salinity, and temperature, to ensure they serve as inputs for the next
time step of the model.

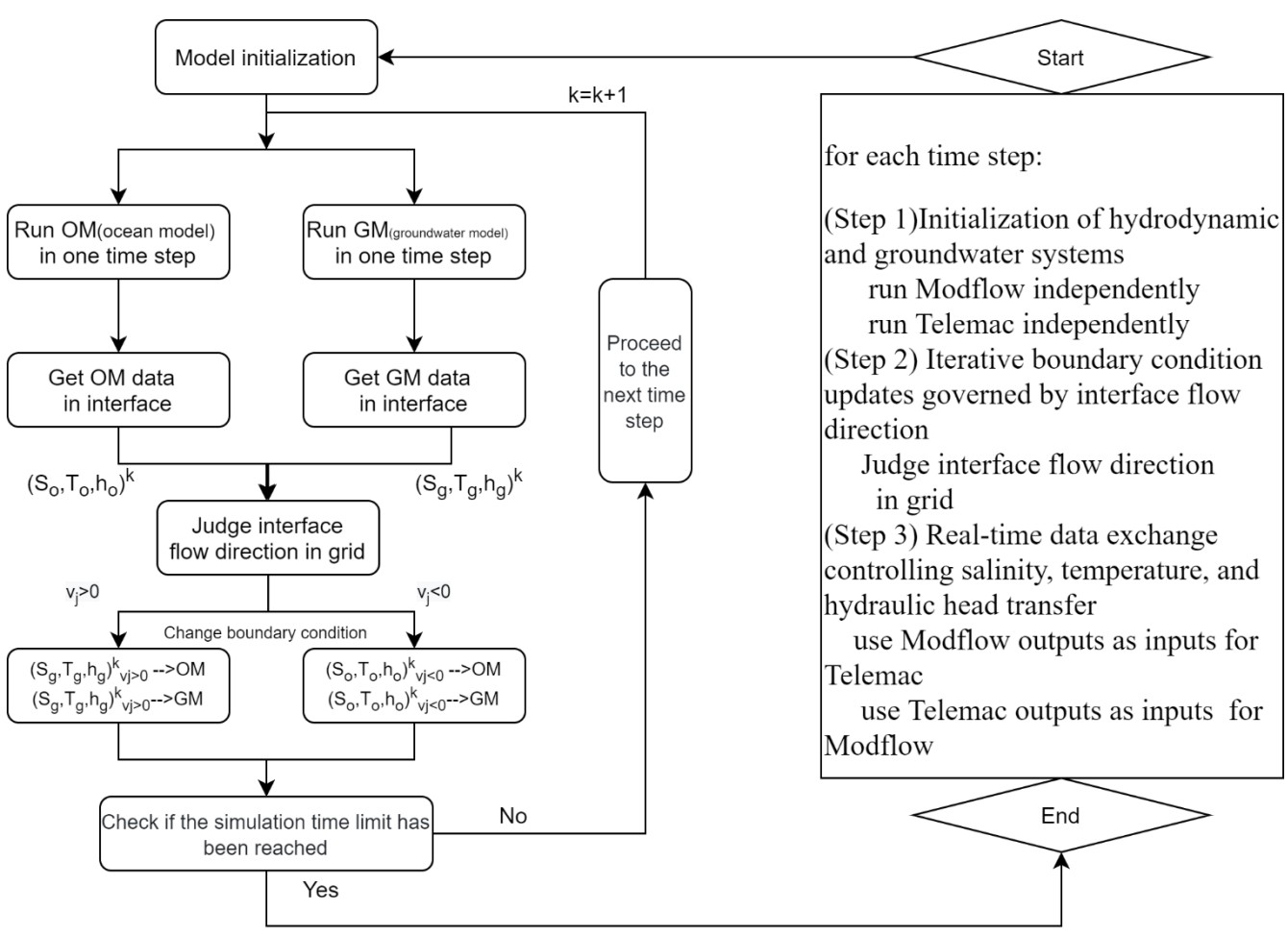

**Fig. 1 Flowchart and Pseudo-code Illustrating the Process of Coupled Models**

**2.3.2 Synchronized Time-Step Coupling**

We built on the temporal coupling framework developed by Yuan et al. (2011) for surface water-groundwater interactions
and implemented a time-step coupling scheme to model how marine dynamics and groundwater systems interact using Telemac
and MODFLOW. In the coupling tidal case, the Telemac model operated at a high frequency with a time step of 10 seconds,
essential for capturing the rapid fluctuations of tides, particularly critical for accurately reproducing the hydrodynamics features

in regions where tidal energy is concentrated. Data, including key hydrodynamic variables such as water levels and flow velocities, were outputted by the Telemac model every 600 seconds (every 10 minutes), serving as inputs to update the dynamic boundary conditions of the coastal region in the MODFLOW model. Despite traditionally favoring longer time steps, the update frequency of the MODFLOW model was specifically tuned within our coupling framework to match the output frequency of the Telemac model, ensuring that the groundwater model could respond in real-time to changes in marine dynamics, accurately reflecting the immediate effects of tides on groundwater resources. Throughout the three-day simulation period, the coupled modeling process iterated continuously: the Telemac model independently ran with a 10-second time step to simulate oceanic processes, transferring its latest dataset to the MODFLOW model every 600 seconds, which then promptly updated its coastal boundary conditions and continued simulating groundwater flow, maintaining dynamic coordination and real-time information exchange between the two models. Time steps and data exchange frequencies were optimized through iterative sensitivity analysis and repeated testing, ensuring high synchronization between models while minimizing errors and maintaining computational efficiency. This approach achieved a balance between simulation accuracy and efficiency

### 2.3.3 Integration of Salinity-Driven Hydrostatic Pressure

In our research, the accurate setting of water head conditions at the land-sea interface has become a focal point, which is critical for the precision of coupled ocean and groundwater models. Given that ocean salinity directly influences seawater density(Fofonoff & Millard Jr, 1983), which in turn affects hydrostatic pressure, we have developed a comprehensive method to obtain vertical salinity profiles at key coastal boundary locations. By obtaining detailed measurements of salinity at various depths, we can calculate the density of seawater at each level, leading to precise estimations of corresponding hydrostatic pressures. This measure is particularly crucial for simulating hydrological processes in coastal areas as it ensures that the boundary conditions received by the groundwater model accurately reflect real-world hydrostatic pressures, rather than simply water depth.

Hydraulic head coupling is implemented bidirectionally. TELEMAC's tidal water level (htide) is converted to a density-corrected head for MODFLOW's General Head Boundary (GHB):

$$h_{GHB} = h_{tide} + \frac{\rho_{sw} - \rho_0}{\rho_0} \cdot z_{bed} \quad (7)$$

where $\rho_{sw} = \rho_0(1 + \beta_S(S_{ocean} - S_0))$ accounts for salinity-driven hydrostatic pressure. Conversely, MODFLOW's groundwater head (hGW) is integrated into TELEMAC's bottom boundary as a pressure flux:

$$P_{bot} = \rho_{GW} \cdot g \cdot h_{GW} \quad (8)$$

where $\rho_{GW}$ depends on groundwater salinity and temperature.

he benefits of adopting this method include enhanced simulation accuracy of groundwater models in coastal regions, allowing them to depict the interactions more realistically between groundwater and the sea, which is essential for understanding coastal hydrological processes. This approach improves the model's resilience to complex changes in marine environments, especially in estuarine zones where salinity gradients are pronounced (Dias et al., 2021), capturing finer details of hydrodynamic characteristics.

## 3 Model construction, validation and application

This section evaluates the coupled model through two scenarios: (1) a laboratory-scale validation against controlled seawater intrusion experiments (Na et al., 2019), and (2) a field-scale application simulating tidal-driven SGD and seawater intrusion. Results are compared to single model to isolate coupling effects.

### 3.1. Validation of the coupled ocean-groundwater model.

### 3.1.1. Laboratory Groundwater model construction

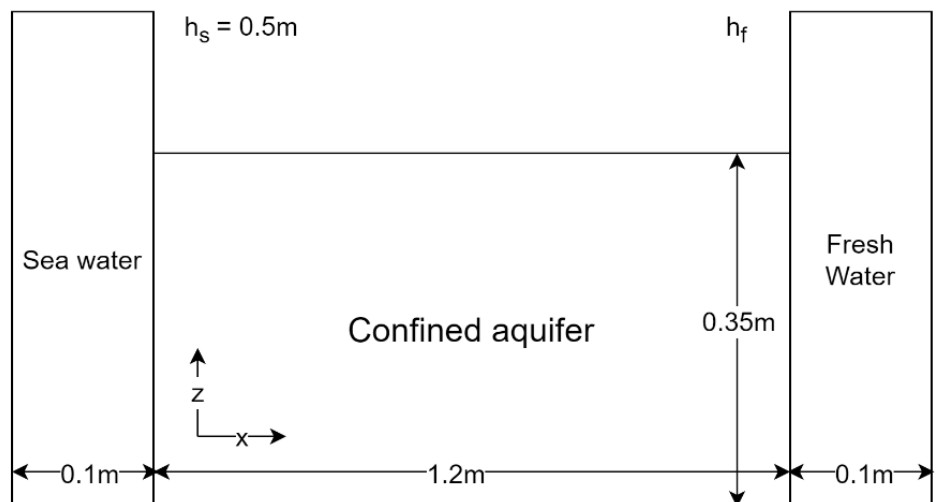

**Fig. 2 Schematic of the coupled ocean-groundwater model: Ocean model (TELEMAC-3D) simulating seawater, groundwater model (MODFLOW6) representing a confined aquifer, with no-flow and constant freshwater head boundary conditions.**

Figure 2 illustrates the schematic diagram of the coupled numerical model, where the ocean component is depicted on the left, the confined aquifer is in the middle, and the freshwater head is shown on the right. The model is used to study the effect of changes in the freshwater head on seawater intrusion. A simplified hypothetical three-dimensional model was constructed based on the confined aquifer at the center. The aquifer model extends 1.2 meters in length and is 0.35 meters thick, discretized into approximately 4000 rectangular elements. The model employs no flow and no transport boundary conditions at the top and bottom, while the freshwater boundary (right boundary) is set as a constant head boundary. The coastal boundary (left boundary) is configured as a general head boundary, utilizing the CHD (Constant Head) and GHB (General Head Boundary) subroutines(Bakker et al., 2016) for defining constant and mixed boundaries, respectively. This setup, common in saltwater intrusion models (Voss & Souza, 1987), allows for the formation of a freshwater outflow zone above the saltwater recirculation region. Comprehensive details on the model parameters can be found in the work of (Na et al., 2019). The model runs for a total of 600 minutes, with a time interval consistent at 10 seconds.

### 3.1.2. Laboratory Ocean model construction

In the early stage of the research, based on the case of studying the impact of seawater density changes on SWI in the laboratory(Na et al., 2019), we constructed a coupling model of ocean and groundwater.

For this study, the ocean model is configured adjacent to the groundwater model, with the ocean's surface set at 0.5m and the bottom at 0m, establishing a constant head boundary condition. The model domain incorporates constant head boundaries with specified salinity and temperature inputs on the left side, reflecting the influence of marine inflows. Salinity is fixed at 35 ppt to represent marine conditions, while temperature follows a constant 20 degrees.

Based on the ocean component depicted on the left side of Fig 2, we constructed a simplified three-dimensional ocean model. The ocean model has a length of 0.1 meters and a depth of 0.5 meters, which is discretized into approximately 2,500 triangular

elements. The top of the model represents a free sea surface, and since the model is set to have a constant sea level, the left and right sides are assigned constant heads, while the upper and lower boundaries are set as fixed interfaces. The model runs for a total of 600 minutes, with a time interval consistent with the groundwater model at 10 seconds.

### 3.1.3. Result of laboratory coupled ocean-groundwater model.

We chose published laboratory experiments on seawater intrusion as the basis for validation(Na et al., 2019), given that the validation cases for the coupled model did not provide information on the marine component. Therefore, we focused on the groundwater model component during the validation process. These experiments meticulously recorded key parameters during the seawater intrusion process, such as salinity distribution and water level changes. We ran the coupled model using the same initial and boundary conditions as those specified in the selected case studies and compared our model output with the experimental data to assess the accuracy and reliability of the coupled model.

We compared the laboratory results figures, the numerical simulation figures published in the article, and the groundwater results from the coupled model. The validation was conducted by rigorously comparing the seawater toe location, the seawater height, and the time it takes for the model to reach a steady state.

Figure 3 shows that the coupled model successfully simulated the transient evolution of the saltwater wedge over time in a 600-minute simulation when the water head height at the land boundary was 0.52 meters. The simulation results are relatively consistent with the laboratory observation data. This practical operation not only verifies the technical feasibility of the new coupled model, but also indicates that it can accurately reflect the actual situation.

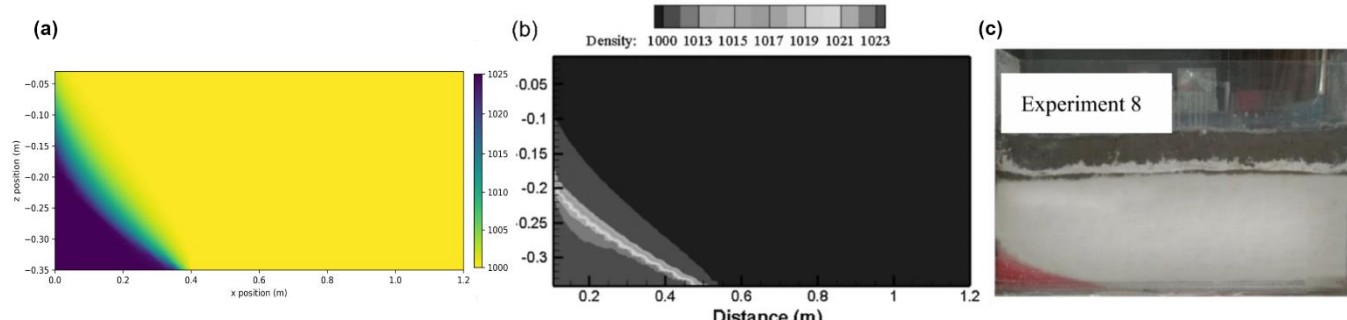

**Fig.3 (a) Coupling model results in Groundwater part (b) Numerical model result in (Na et al., 2019) (c) Laboratory simulation results**

In this study, the CellBudgetFile module in Flopy was utilized to process the flux and flow direction of boundary units in the groundwater model at each time point during the entire simulation process. A comparison of the simulated data for Submarine Groundwater Discharge (SGD) and Recirculated Submarine Groundwater Discharge (RSGD) across different time periods in the two models reveals significant differences. RSGD is distinct from the broader surface water-groundwater interactions described by hypokymatic flow(Wilson et al., 2016). In our classification, RSGD is categorized under saline SGD, which refers to discharge with salinity greater than 5 ppt.

In terms of SGD simulation, the coupled model calculates an SGD value of $3.6e^{-2}$ m³, slightly higher than the $3.4e^{-2}$ m³ calculated by the single model (see Table 1). This suggests that the coupled model more accurately reflects the complex material exchange between seawater and groundwater, thereby avoiding potential underestimation. For RSGD, the coupled model yields a value of $7.5e^{-3}$ m³, which is up to 17% higher than the $6.4e^{-3}$ m³ obtained from the single model. This difference highlights the coupled model's superior analytical capability and higher precision in capturing the saltwater recirculation mechanism.

**Table 1. Differences between the coupled model and the single groundwater model in simulating SGD.**

| Unit: $e^{-2}$m³ | SGD | FSGD | RSGD |
|---|---|---|---|
| coupling model | 3.6 | 2.8 | 0.8 |
| single model | 3.4 | 2.8 | 0.6 |

Standalone Groundwater Validation: The MODFLOW6 module was validated against Na et al.'s (2019) laboratory data,

replicating seawater intrusion (SWI) with <5% error in toe position (0.38 m simulated vs. 0.36 m observed).Coupled System

Test: A simplified TELEMAC domain (0.1×0.5 m) was added to Na et al.'s setup. The coupled model maintained SWI

accuracy (0.38 m toe position) while capturing 17% higher RSGD. This confirms that discrepancies arise from realistic

coupling, not model errors.

**Table 2. Comparison of coupled/single models and laboratory experiments on SWI toe position and SWI height**

| Parameter | Laboratory Data | Na numerical | Coupled model | Single model |
|---|---|---|---|---|
| SWI toe (m) | 0.36 | 0.41 | 0.38 | 0.38 |
| SWI height (m) | 0.15 | 0.17 | 0.17 | 0.18 |

Traditional single models dealing with seawater intrusion (SWI) typically impose a constant salinity boundary condition,

assuming that the salinity at the seawater-groundwater interface remains unchanged. This simplification overlooks the fact that

freshwater discharge from the aquifer in real-world environments can reduce salinity at the sea-land interface, potentially leading

to an underestimation of the effective discharge volume of SGD.

In contrast, the coupled model employs dynamic boundary conditions, which overcome this limitation. Within the

framework of the coupled model, the boundary is no longer a static salinity barrier but becomes a responsive interface that adjusts

with time and environmental conditions. When freshwater is discharged into the ocean through the SGD process, the dynamic

boundary can immediately reflect the decrease in seawater salinity. Conversely, during the RSGD process, seawater back-

infiltration can increase groundwater salinity, and these changes are accurately captured by the model. This approach allows the

model to not only simulate salinity distribution more accurately but also to reflect the natural fluctuations in water-salt exchange

in coastal waters, thereby improving the realism of the simulation and the accuracy of SGD and RSGD quantification.

At the same time, the bidirectional feedback mechanism implemented in the coupled model further enhances this advantage.

It ensures that the interaction between the ocean and the groundwater flow system is not unidirectional, but an interactive process.

For example, changes in ocean tides not only affect the position and salinity of the seawater boundary, but also regulate the flow

pattern of groundwater; and the rise and fall of groundwater levels can also feedback affect the hydrodynamic status of the

nearshore marine area.

Although the improvement of the coupled model is not prominent in the basic case, when we introduce the coupled model

into dynamic tides, due to its use of dynamic boundaries and bidirectional feedback mechanisms, the coupled model can help to

understand those complex hydrogeochemical processes that are difficult for independent models to reveal.

**3.2. Ocean-groundwater model with Tidal Boundary.**

**3.2.1. Ocean-groundwater model construction**

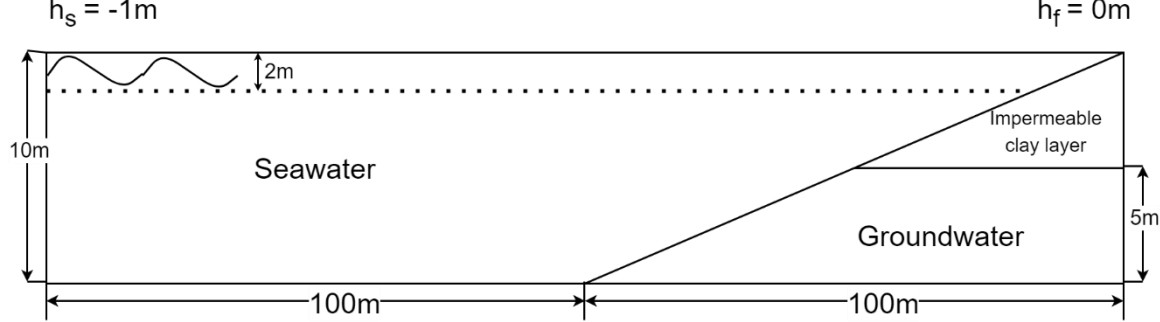

**Fig. 4 Initial and boundary conditions of the tide numerical model**

Figure 4 presents a coupled numerical model schematic under tidal boundary conditions, with a sloping ocean on the left

side and a confined aquifer on the right side. The mean sea level set at -1 m while tidal fluctuation ranges from -2 m (low tide)

to 0 m (high tide). The triangular region above the confined aquifer represents an impermeable clay layer, preventing vertical flow. This coupled model aims to investigate the dynamics of submarine groundwater discharge and seawater intrusion under tidal influences.

The groundwater model component is a simplified three-dimensional structure with dimensions of 100 meters (length) × 50 meters (width) × 5 meters (thickness), discretized into approximately 4000 rectangular elements. No-flow boundary conditions are applied at the top and bottom of the model. The freshwater boundary (right boundary) is set as a constant-head boundary at 0 meters with a water temperature of 10 degrees Celsius. The coastal boundary (left side) is set as a general head boundary condition, defined using the Constant Head (CHD) and General Head Boundary (GHB) subroutines. The basic parameters used in the current simulation are shown in Table 3.

**Table 3. Parameters used in the numerical simulations of the groundwater part**

| Parameter | Value |
| --- | --- |
| Aquifer porosity | 0.35 |
| Hydraulic conductivity | 5e-3m/s |
| Saltwater density | 1025 kg/m$^3$ |
| Saltwater elevation | -1m |
| Freshwater density | 1000 kg/m$^3$ |
| Freshwater elevation | 0m |
| The coefficient of molecular diffusion | 1e-9m$^2$/s |

The ocean model component utilizes the TELEMAC system, with dimensions of 200 meters (length) × 50 meters (width) × 10 meters (depth) and is discretized into approximately 2700 triangular elements. The top of the model represents the free surface of the sea, where tidal variations are introduced with a range from -2 meters to 0 meters. The seabed is set at -10 meters. The left boundary of the domain is specified with tidal boundary conditions representing the tidal water level, with salinity set to a constant 35 ppt, and temperature varying sinusoidally from 20 to 25 degrees Celsius follows a sinusoidal cycle with each tidal state to reflect the influence of marine inflow. The right boundary serves as a fixed boundary, representing the actual coastline, while still allowing dynamic adjustments based on tidal fluctuations to capture land-sea interactions.

The salinity value of 35 ppt was selected to represent typical conditions in temperate coastal zones (e.g., the Atlantic Ocean near Europe), where field observations and laboratory experiments report salinities of 32–36 ppt (Na et al., 2019; Martínez-Pérez et al., 2022). The sinusoidal temperature variation (20–25°C) simulates diurnal and seasonal thermal fluctuations in shallow coastal waters, consistent with Mediterranean estuaries influenced by solar heating and groundwater discharge (Nguyen et al., 2020).

The model operation is divided into two phases: the first phase (without tidal effects) simulates the system until it reaches a steady state. This initial phase helps to eliminate any potential disturbances from the starting conditions and allows the system to stabilize, establishing a baseline state where external temporal factors do not influence the results. The second phase lasts for 72 hours, with a time step of 10 minutes, and the groundwater model and the ocean model had the same time step. This two-phase approach allows for a clear distinction between transient tidal effects and the underlying groundwater-seawater interactions, ensuring that the model accurately reflects both the steady-state conditions and the dynamic impact of tidal forces.

**3.2.2. Result of ocean-groundwater model**

This study investigates the dynamic interactions between ocean and groundwater systems, particularly focusing on the salinity distribution and its evolution over time as depicted in Fig. 5. The upper part of the figure shows a top-down view of the ocean model, while the lower part presents a side view of the coupled model.

Initially, the model starts with a distinct salinity gradient, where the ocean side is characterized by a uniformly high salinity of 35 (purple), while the adjacent land aquifer is under a pressurized freshwater condition (yellow). This setup creates a sharp boundary between saltwater and freshwater, setting the stage for subsequent interactions.

As the system evolves towards equilibrium (Fig. 5b), a stable salinity distribution emerges, characterized by the formation of a saltwater wedge that extends from the ocean into the aquifer. This wedge structure is indicative of the balance achieved between the ocean's saline water and the freshwater outflow driven by Submarine Groundwater Discharge (SGD). The boundary between the saltwater and freshwater becomes more defined, showing a delicate balance where freshwater from the aquifer spreads over the ocean surface and gradually mixes with seawater.

A complete sinusoidal tidal cycle (b-f) is shown to illustrate the effects of tidal fluctuations on this balance. The impacts of these fluctuations are clearly visible during high tide (Fig. 5c) and low tide (Fig. 5e). During high tide (b-c), the rising sea level drives more saline water into the aquifer, causing the saltwater wedge to expand further inland, as indicated by the increased red area. Conversely, at low tide, the retreating sea level allows some saltwater to flow back into the ocean, reducing the wedge's inland penetration and allowing the freshwater in the aquifer to recover. This cyclical process highlights the significant impact of tidal forces on the salinity dynamics at the sea-land interface.

Additionally, the lag effect of tidal fluctuations on seawater intrusion is clearly demonstrated. During high tide (Fig. 5c), the rising sea level causes the saltwater wedge to rapidly advance inland. However, it can be observed that even though the sea level has started to rise, the maximum extent of the saltwater wedge does not immediately reach its peak but takes some time to fully intrude. Conversely, during low tide (Fig. 5e), when the sea level drops, the saltwater wedge begins to retreat. However, the saltwater does not immediately exit the aquifer; instead, it shows a slow withdrawal trend. Even after the sea level has dropped, the saltwater remains within the aquifer for some time. This phenomenon indicates that despite the rapid changes in tidal levels, the process of seawater intrusion exhibits a certain degree of lag. This reflects that the response speed of the groundwater system is slower than the changes in sea level.

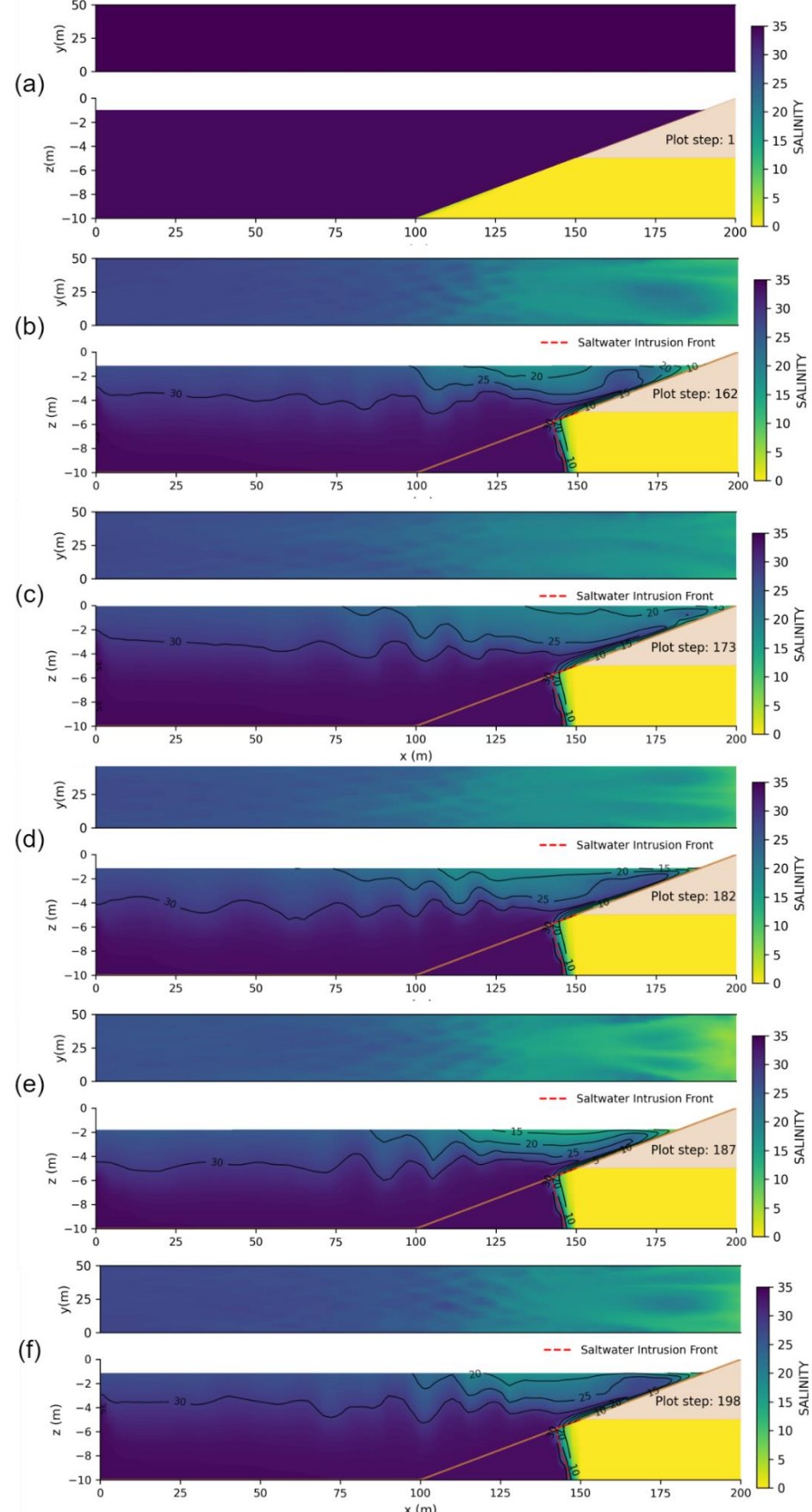

Fig. 5 Coupling model simulation of salinity patterns throughout tidal cycles depicted in top and vertical views

(a)Initial moment (b)Tidal Start (c) High Tide (d)Mid-Tide (Flood) (e) Low Tide (f) Mid-Tide (Ebb)

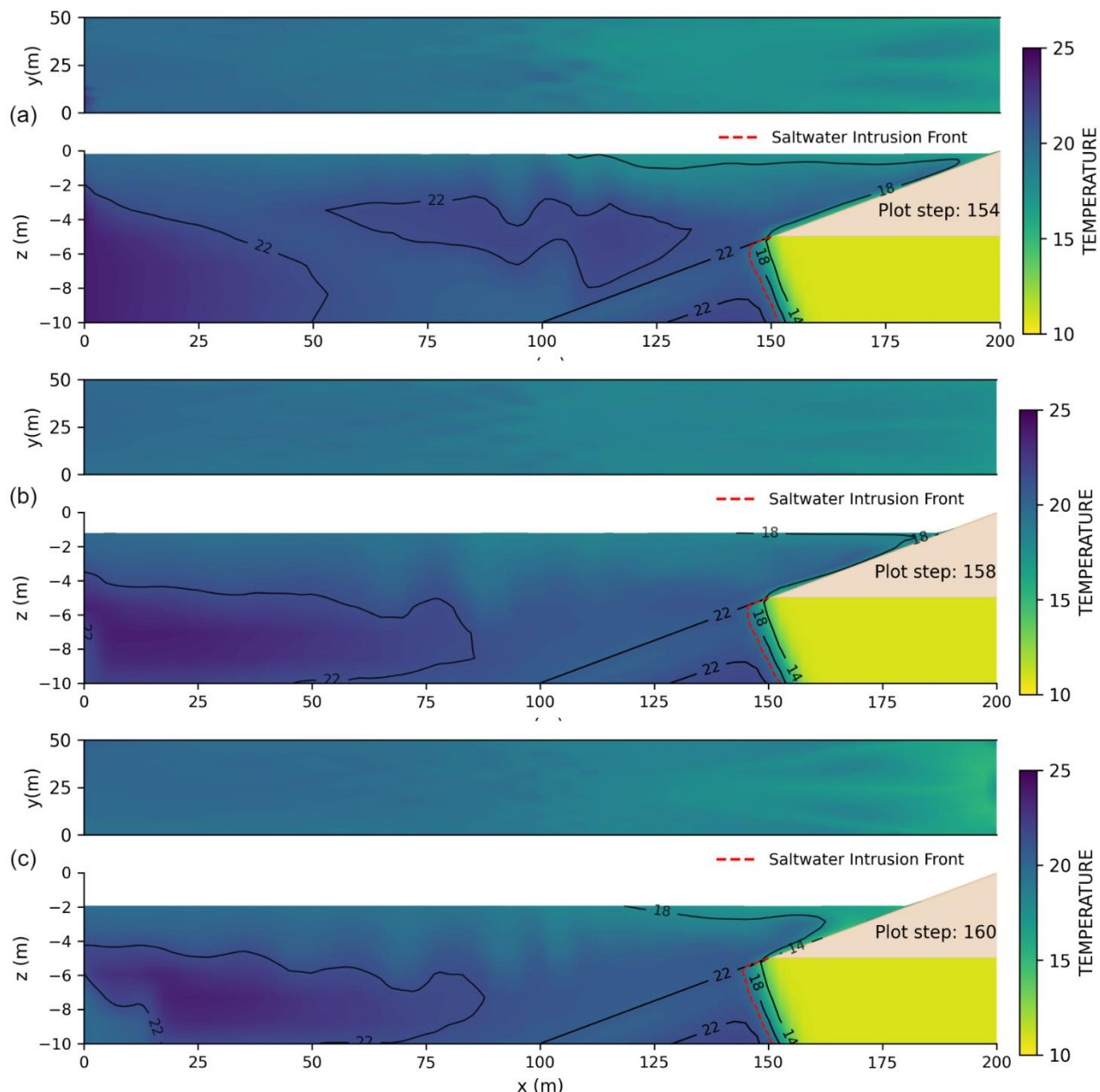

**Fig. 6 Temperature patterns throughout tidal cycles depicted in top and vertical views (a)  High Tide moment (b) Mid-Tide (Ebb) (c)  Low Tide moment**

Like salinity distribution, temperature variations are also influenced by the dynamics of groundwater discharge (SGD). Specifically, temperature changes in the upper layer of the ocean are observed due to SGD. Fig 6 illustrates three distinct time points, reflecting the impact of submarine groundwater discharge on ocean temperatures as the tide transitions from high to low.

Figure 6a depicts the scenario when the tide is at its highest point. At this stage, cooler water from the aquifer spreads along the land-sea interface to the ocean surface, creating a distinct cold front that is particularly evident in the upper layers of the sea. With the tide at its peak, there is maximum seawater coverage, and a large volume of warmer seawater enters from the left side, resulting in a concentrated temperature distribution with a noticeable temperature gradient.

As the tide begins to recede but has not yet reached its lowest point (Fig. 6b), the colder groundwater discharged is gradually extending outward due to the decreasing tide, mixing with the surrounding warmer seawater. This leads to a more uniform temperature distribution and a reduction in the temperature gradient.

Figure 6c describes the situation when the tide reaches its lowest point. At this time, the seawater covers the smallest area,
allowing the colder groundwater discharged to accumulate more easily on the ocean surface, awaiting the next tidal cycle for
further mixing.  This simulates real-world coastal temperature and salinity dynamics, where tidal cycles influence the layering
of water temperature.

**3.2.3. Comparison of the ocean-groundwater model with single model**

To comprehensively assess the advantages of the coupled model, this study also separately ran the standalone ocean model
and groundwater model. In the ocean model, a specified flux boundary condition was set at the top of the interface between the
ocean and the confined aquifer to simulate submarine groundwater discharge. In the groundwater model, a general head boundary
(GHB) condition with a fixed salinity of 35 was applied at the land-sea interface to simulate interactions between seawater and
groundwater, where the continuous hydraulic gradient drives seepage, causing freshwater to discharge through the aquifer to the
seabed. By comparing the simulation results of the single model and the coupled model (Fig. 7), it becomes clearer that the
coupled model can more effectively capture the dynamic changes in complex environments.

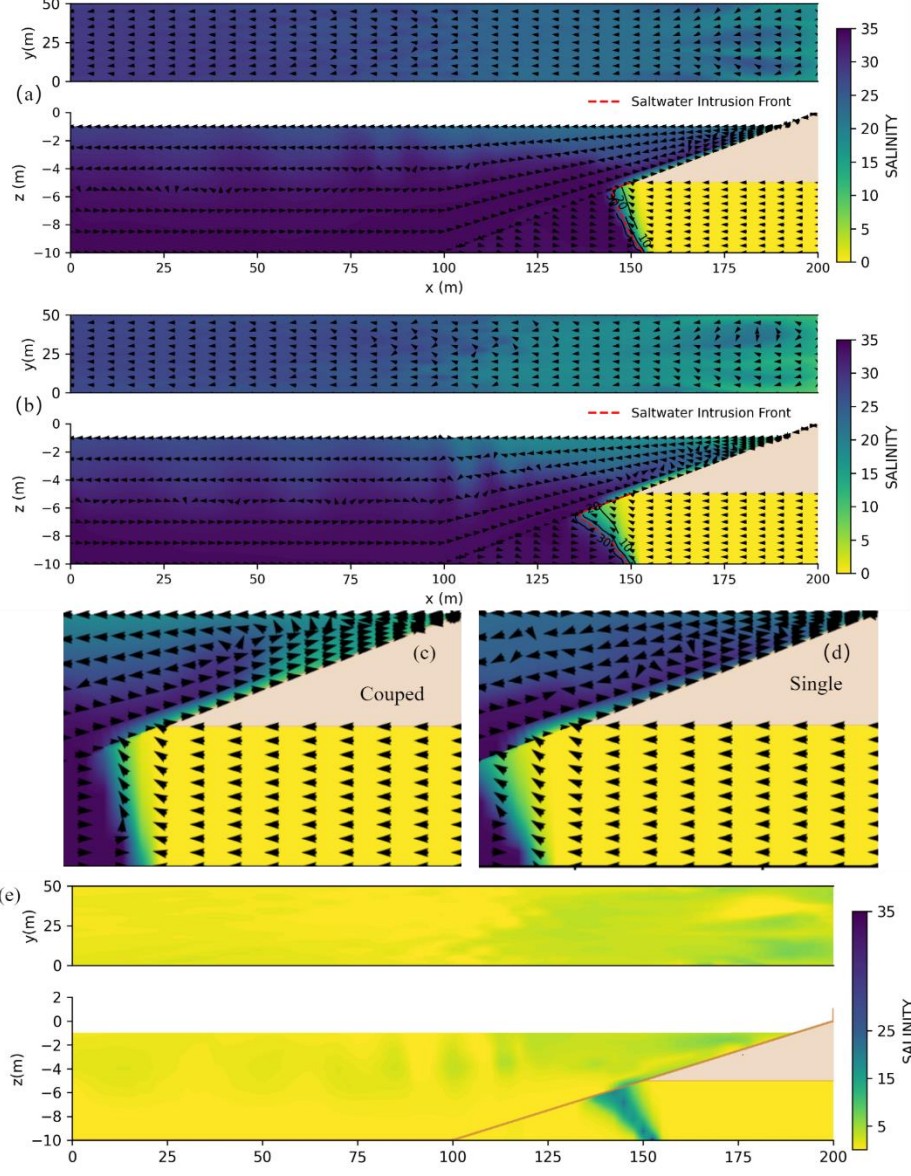

**Fig. 7 Comparison of salinity and velocity  between two models from top and vertical views (a) Coupled Model (b)**
**Single Model  (c) (d) Enlarged detail image (e) Coupled – Single difference image.**

In the ocean top-view of Fig 7, dynamic variations caused by the interaction between the ocean surface and groundwater are captured in the coupled model. The arrows on the ocean surface in the coupled model, which indicate flow direction and velocity, appear somewhat disordered. This is due to the complex local flows generated by groundwater discharge in the coupled model. Unlike a conventional ocean model, the coupled model integrates dynamic submarine groundwater discharge (SGD), which induces localized convection, eddies, and small-scale mixing effects near the seafloor. This phenomenon indicates that the coupled model more accurately represents the complex conditions found in real-world environments.

The observed patterns in the figure suggest that regions with disordered arrows extend towards the ocean due to groundwater discharge. In the coupled model, the discharge area is confined to the upper portion of the land-sea interface, where complex interactions between upwelling groundwater and seawater create multiple turbulence zones. Within these areas, flow exhibits irregular and rapidly fluctuating patterns. The extension of disordered arrow regions toward the ocean implies that groundwater discharge not only introduces a new mechanism for material transport into the ocean but also provides additional kinetic energy, thereby enhancing mixing processes in nearshore waters.

In the view on the right side of Fig 7, it is evident that more freshwater accumulates at the ocean surface in the coupled model compared to the standalone model. This is due to the faster discharge rate and higher volume of groundwater discharge in the coupled model.

In the cross-sectional view of groundwater, Fig 7a shows the shape of seawater intrusion in the coupled model, while Fig 7b presents the shape in the single groundwater model, which is comparatively more gradual than that of the coupled model. Using the CellBudgetFile module, we obtained groundwater flow velocities at the submarine discharge outlet for both models. Results show that the discharge velocity in the coupled model is 30% higher than in the single model. To better show the difference between the two models, Figure 7(e) displays the salinity difference by subtracting the standalone model's salinity from the coupled model's salinity.

In the single model, the land-sea boundary salinity is fixed at a relatively high level, resulting in higher pressure heads at the boundary. This reduces the head difference with inland areas and slows groundwater flow velocity. This slower groundwater discharge effectively resists seawater intrusion, resulting in a more gradual saline wedge shape with a shallower intrusion depth. In contrast, the coupled model incorporates a gradation of boundary salinity from the surface to the bottom, with lower salinity levels in the upper zones and lower pressure heads. This leads to faster groundwater flow. The increased discharge rate decreases freshwater pressure and creates a head difference that encourages seawater to advance inland to fill the gap created by freshwater discharge. This pressure differential accelerates seawater intrusion into the aquifer, resulting in a steeper saline wedge and deeper intrusion depth.

Comparing the shapes of seawater intrusion between the single and coupled models demonstrates that boundary conditions and discharge rates significantly impact seawater intrusion patterns. The coupled model provides a more accurate representation of the actual hydrodynamic conditions at the land-sea interface, offering a more precise perspective for understanding seawater intrusion mechanisms.

### 3.2.4. The Impact of Tides on SGD

The influence of tides on submarine groundwater discharge (SGD) is significant, characterized by pronounced periodic variations(Li et al., 2016). Specifically, during high tide, the increased seawater pressure not only suppresses SGD but may even cause seawater to flow back into the aquifer, exacerbating seawater intrusion. Conversely, during low tide, as seawater pressure decreases, the discharge of groundwater into the ocean significantly increases, indicating that SGD exhibits non-steady-state characteristics(Fang et al., 2021b).

In the comparison between the coupled model and the single model, we distinguish the types of submarine groundwater discharge based on salinity. Due to the mixing of freshwater and brackish water at the land-sea interface, we simplify the classification by considering water with salinity less than 5 as freshwater. Our findings show that during a 24-hour steady-state observation and a 72-hour tidal fluctuation period, as illustrated in Table 4, the SGD for the coupled model is 97.5 m³, representing a 17% increase compared to the single model's 83.1 m³. The total amount of freshwater groundwater discharge (FSGD) is

comparable between the two models. However, the recirculated submarine groundwater discharge (RSGD) in the coupled model is 39.4 m³, a 54% increase over the single model's 25.6 m³. This indicates that RSGD contributes almost entirely to the increase in SGD.

**Table 4. Analysis of Submarine Groundwater Discharge (SGD) in Coupled and Single Models.**

| unit:m³ | Coupled model | | | Single model | | |
|---|---|---|---|---|---|---|
| SGD | FSGD | RSGD | SGD | FSGD | RSGD | |
| 97.50 | 58.10 | 39.40 | 83.10 | 57.50 | 25.60 | |

Tidal fluctuations can significantly alter the ratio of RSGD to FSGD. As shown in Table 4, under tidal influence, the performances of SGD in coupled models and single models exhibit significant dynamic variations, with the coupled model providing a more accurate reflection of the tidal-induced groundwater flow characteristics. Under steady-state conditions, the SGD values for the coupled model and single model are $25.50 \times 10^{-5}$ m³/s and $22.90 \times 10^{-5}$ m³/s, respectively. In the coupled model, FSGD and RSGD are $16.80 \times 10^{-5}$ m³/s and $8.70 \times 10^{-5}$ m³/s, while in the single model, they are $16.70 \times 10^{-5}$ m³/s and $6.20 \times 10^{-5}$ m³/s, respectively. These results indicate that under steady-state conditions, the coupled model captures more recirculated water flow, demonstrating a closer exchange with oceanic water.

**Table 5. Dynamic Variations of submarine groundwater discharge (SGD) Under Tidal Influence between Coupled model and Single model.**

| unit:$e^{-5}$m³/s | Coupled model | | | Single model | | |
|---|---|---|---|---|---|---|
| | SGD | FSGD | RSGD | SGD | FSGD | RSGD |
| Steady state | 25.50 | 16.80 | 8.70 | 22.90 | 16.70 | 6.20 |
| High tide | 16.70 | 12.20 | 4.50 | 5.20 | 5.10 | 0.10 |
| Middle tide | 28.00 | 17.40 | 10.60 | 26.10 | 18.20 | 7.90 |
| Low tide | 46.20 | 21.30 | 24.90 | 46.40 | 28.10 | 18.30 |

Under the influence of tidal fluctuations, SGD, FSGD, and RSGD in both coupled and single models exhibit periodic changes. At high tide, due to the increase in seawater pressure, the SGD for the coupled model and the single model decrease to $16.70 \times 10^{-5}$ m³/s and $5.20 \times 10^{-5}$ m³/s, respectively. During this period, RSGD significantly declines, with values of $4.50 \times 10^{-5}$ m³/s in the coupled model and only $0.10 \times 10^{-5}$ m³/s in the single model. This phenomenon suggests that seawater pressure suppresses groundwater discharge at high tide, reducing the flow of recirculated water and allowing freshwater discharge to dominate.

As tidal levels decline, SGD gradually increases, reaching its maximum value at low tide, where the SGD in the coupled model and single model rises to $46.20 \times 10^{-5}$ m³/s and $46.40 \times 10^{-5}$ m³/s, respectively. During low tide, RSGD and FSGD in the coupled model peak at $24.90 \times 10^{-5}$ m³/s and $21.30 \times 10^{-5}$ m³/s, respectively, with RSGD slightly exceeding FSGD. This indicates that the decrease in tidal levels facilitates greater recirculation of seawater into the aquifer, enhancing the discharge of recirculated water.

Overall, tidal effects significantly influence the total amount of SGD and the ratio of FSGD to RSGD, particularly evident during high and low tide. The coupled model, by incorporating the dynamic processes of oceanic and groundwater interactions, can more accurately reflect the changes in groundwater discharge driven by tidal forces, including the suppression effects during high tide and the enhancement effects during low tide. Compared to the single model, the coupled model is more representative of the real ocean-groundwater interactions, exhibiting more complex discharge dynamics, thereby contributing to a deeper understanding of the mechanisms behind tidal-driven submarine groundwater discharge.

The coupled model shows an 11.3% increase in total SGD ($25.50 \times 10^{-5}$ vs. $22.90 \times 10^{-5}$ m³/s) and 40.3% higher RSGD ($8.7 \times 10^{-5}$ vs. $6.20 \times 10^{-5}$ m³/s) compared to the single model (Table 5). These differences exceed typical uncertainty thresholds (10%) for SGD-driven nutrient fluxes, justifying the need for coupling in dynamic tidal zones. However, in steady-state systems with minimal oceanic variability, simpler uncoupled models may suffice.

# 4 Conclusion

This study innovatively addresses the long-standing challenge of independently modeling groundwater and ocean systems by implementing a coupled framework that effectively exchanges data at the land-sea interface. This research led us to formulate the following pivotal conclusions:

1. This study has developed a coupled model for simulating the interaction between groundwater and ocean systems. The model's potential and accuracy have been preliminarily validated through experimental case studies, demonstrating its effectiveness in this complex environmental context. The coupled model successfully simulated the transient evolution of the saltwater wedge over time, showing relative consistency with laboratory observation data, thereby verifying the technical feasibility and accuracy of the model in reflecting actual conditions.

2. The coupled model developed in this study demonstrates significant advantages in simulating the interaction between groundwater and ocean systems, particularly in dealing with dynamic boundary conditions and mixing zones in coastal areas. Compared with traditional single groundwater models, the coupled model not only captures fluctuations in salinity and temperature but also specifically considers the effects of mixing zones that are often overlooked. This integrated simulation approach allows for more precise definition of boundary conditions and realistically reflects the hydrogeological and geochemical processes at the interface, thereby enhancing the accuracy and comprehensiveness of the simulation. The coupled model's dynamic boundary conditions and bidirectional feedback mechanisms provide a more accurate reflection of the complex material exchange between seawater and groundwater, avoiding potential underestimation of effective discharge volumes.

3. The coupled model employed in this study vividly illustrates the dispersion of Submarine Groundwater Discharge (SGD) in the marine environment, offering a novel perspective on the interactions at the groundwater-ocean interface. Tidal fluctuations were found to significantly influence the rate and pattern of SGD, thereby modulating the input of nutrients and potential contaminants into the ocean, and revealed the intricate mechanisms of their diffusion, transformation, and accumulation within marine ecosystems. Moreover, the dynamic response of the ocean has a substantial impact on the pathways and spatial distribution of SGD, underscoring the necessity of incorporating an oceanographic viewpoint in studies related to SGD. The coupled model's ability to simulate the effects of tidal influences on SGD and seawater intrusion provides a more accurate understanding of these complex hydrogeochemical processes.

To build upon the findings of this study and further advance our understanding of the complex interactions between groundwater and ocean systems, the following research directions are proposed:

1).Explore how variations in bathymetry and tidal patterns, as well as different hydraulic conductivities, affect the dynamics of seawater intrusion (SWI) and submarine groundwater discharge (SGD). Additionally, investigate the impact of temperature and salinity changes on the movement and dispersion of SGD in coastal environments.

2).Integrate biogeochemical processes into the coupled model to simulate the chemical transformations that occur as SGD interacts with the ocean. Develop reactive transport models that consider the reactions between groundwater constituents and the marine environment.

3).Apply the coupled model to real-world coastal systems to validate the model against observed data. Deploy additional observational points in both terrestrial and marine environments to collect critical data for model calibration and validation. Conduct scenario analysis to predict the impacts of climate change, such as rising sea levels and increased storm events, on SGD and SWI.

4) Future studies should integrate porewater flow dynamics (e.g., wave-driven advection) to resolve benthic exchange processes, which are essential for understanding coastal carbon and nutrient cycles.

5) Parameter sensitivity analyses reveal that salinity and temperature gradients significantly influence coastal dynamics. These findings highlight the need for site-specific parameter calibration in future applications.

**Code/Data availability**

Some or all data, models, or code generated or used during the study are available from the corresponding author by request.

**Author contribution**

Jiangyue Jin (First Author): Conceptualization, Methodology, Code, Writing - Original Draft;

Manuel Espino: Theoretical guidance for the ocean model; Supervision, editing & revising the manuscript;

Daniel Fernández: Theoretical guidance for the groundwater model; editing & revising the manuscript

Albert Folch (Corresponding Author): Theoretical guidance for the groundwater model, Funding Acquisition, Resources, Supervision, Review & Editing.

**Competing interests**

The authors declare that they have no conflict of interest.

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
