# Peer review of "Coupling of numerical groundwater-ocean models to improve understanding of the coastal zone"

_EGUsphere, 2024_

## Author Response (AR1)

Thank you for your answer and the reviewers' comments on our manuscript, "Coupling of numerical groundwater-ocean models to improve understanding of the coastal zone" (ID: EGUSPHERE-2024-3384). We sincerely appreciate the valuable and insightful feedback, which has help to significantly improve our paper. We have carefully considered all the comments and made the necessary revisions, which we hope will meet your approval.

**This manuscript presents a new model that dynamically couples a three-dimensional hydrodynamic model with a groundwater model to capture ocean-land interaction related to groundwater discharge and seawater intrusion. The authors simulate groundwater and ocean water dynamics for a laboratory setup and a realistic coastal setup, both fairly classical scenarios. Simulation results show enhanced submarine groundwater discharge and seawater intrusion when interaction is present. The development of such a model is both challenging and highly relevant to researchers in both oceanography and groundwater studies. I find the manuscript suitable for publication, after some clarifications.**

We really appreciate the reviewer highlighting the challenge and the importance of the work submitted to this journal.

**General comments 1**
**The description of the model in Section 2 requires improvement, particularly on the boundary conditions (BC) used to couple the two models, as this is the central part of the manuscript. Most of the coupling methodology described in Section 2.3 (Step 2) addresses how the model components are engineered and how the interface is set up (Fig. 1). Subsection 2.3.2 primarily discusses the BC for MODFLOW and not TELEMAC. It would be helpful to explicitly state the actual BC used in equation form, enabling researchers to apply similar methods to other numerical models within the community. Key aspects that need clarification include: What specific parameters are exchanged at the boundary, and at what dynamic conditions? Is there a mass flux across the interface? Are salinity and temperature fluxes considered at the boundary based on mass flux but without accompanying mass input? Or are the profiles specified at the boundary and diffused? How is the hydraulic head incorporated into the coupling? The descriptions on P5 L28-29 and P6 L1-7 are vague and need to be elaborated to address these points in detail.**

**Response to general comment 1:**
Thank you for your feedback regarding the need for clearer descriptions of the coupling methodology, particularly the boundary conditions (BC) between the TELEMAC and MODFLOW models. We have revised the manuscript accordingly to address your concerns in detail and improve clearness. Below, we provide a point-by-point response to your questions:
**1.What specific parameters are exchanged at the boundary, and at what dynamic**

**conditions?**

Your observation about the need for explicit mathematical formulations and a more balanced description of both models' BCs is well-founded, as these details are indeed critical for reproducibility and broader application. In revising Section 2.3, we recognize that our initial manuscript did not fully articulate the dynamic parameter exchange rules or the specific BC implementations for TELEMAC, which may have hindered clarity.

To address this, we will restructure the coupling methodology to explicitly define the exchanged parameters (hydraulic head, salinity, and temperature) and their dynamic interactions in equation form (Equations 5–6). Section 2.3.1 (Page 7, Lines 21–27)

The GWT module of Modflow6 simulates temperature and salt transport in groundwater by solving the Advection-Dispersion Equation (ADE):

$\partial(\theta C)/\partial t = \nabla\cdot(\theta D\nabla C) - \nabla\cdot(qC) + Qs,$

where C represents concentration (salinity) or temperature, $\theta$ is porosity, D the dispersion tensor, q Darcy velocity, and Qs source/sink terms.

TELEMAC-3D resolves marine mixing through 3D Navier-Stokes equations coupled with turbulent diffusion and advection:

$\partial C/\partial t + u\cdot\nabla C = \nabla\cdot(\nu_t\nabla C) + S,$

where $\nu_t$ denotes turbulent diffusivity and S external sources.

For TELEMAC, the bottom pressure boundary now incorporates MODFLOW's groundwater head ($h_{GW}$), while salinity and temperature boundaries are dynamically assigned based on the direction of groundwater flux (Q). Specifically, when Q>0 (groundwater discharge to the ocean), TELEMAC adopts MODFLOW's salinity ($S_{GW}$) and temperature ($T_{GW}$); conversely, during seawater intrusion (Q<0), MODFLOW's boundaries are updated using TELEMAC's outputs (Socean, Tocean). These rules will be illustrated in the updated flowchart (Fig. 1) and integrated into Section 2.3.1 (Page 7 line10-13)in the revised version of the paper. These revisions aim to enhance methodological transparency and help researchers to adapt the approach to other coupled groundwater-surface water systems.

Section 2.3.1 (Page 7, Lines 9–18)
**2.Is there a mass flux across the interface?**
**3.Are salinity and temperature fluxes considered at the boundary based on mass flux but without accompanying mass input?**
**4.Or are the profiles specified at the boundary and diffused?**

We acknowledge that our initial manuscript did not explicitly clarify how salinity and temperature fluxes are quantitatively linked to groundwater-seawater exchange, and we will revise the text to address this gap.

In the current model, explicit mass flux exchange across the interface was not directly implemented. This simplification stems from the fundamental scale disparity between groundwater systems (governed by Darcy-scale velocities) and marine hydrodynamic processes (resolved through Navier-Stokes equations), where direct coupling of mass

fluxes would induce numerical instabilities. The seepage velocity of groundwater typically ranges from $10^{-6}$ to $10^{-4}$ m/s, while oceanic flow velocities (e.g., tidal currents or coastal currents) are generally on the order of $10^{-2}$ to 1 m/s, differing by several orders of magnitude. As a result, the contribution of groundwater to the salinity and temperature mass flux in the ocean is negligible on a per-unit-time basis. Moreover, the high mixing rate at the ocean boundary, dominated by turbulent diffusion, rapidly dilutes the groundwater flux. To overcome this multi-physics coupling challenge, we adopted an interface node approach for salinity/temperature parameter transfer: By monitoring flow direction at interfacial nodes (e.g., land-to-sea discharge or seawater intrusion), unidirectional transport of salinity/temperature gradient data is triggered to the receiving model while maintaining independent mass field evolution in both domains. Although this design does not explicitly incorporate mass fluxes, it implicitly accounts for solute diffusion and heat conduction effects through parameter transfer (e.g., the ocean model updates boundary layer salinity via diffusion terms after receiving groundwater salinity data) simulating the head and salinity exchange of the aquifer and the ocean.

To clarify this point Section 2.3.1 (Page 7, Lines 28–36) has been modified accordingly.

**5.How is the hydraulic head incorporated into the coupling?**
**6.The descriptions on P5 L28-29 and P6 L1-7 are vague and need to be elaborated to address these points in detail.**
We thank the feedback regarding the coupling mechanism clarification. The original manuscript's vague descriptions of hydraulic head coupling in coastal zones (particularly on Pages 5-6) indeed required more mathematical treatment. To address this, we will fundamentally restructure Section 2.3.3 to explicitly articulate the bidirectional coupling logic and its physical implementation. Specifically, the revised text will detail how TELEMAC's tidal water level (htide(t)) undergoes density correction (accounting for seawater salinity variations) before being imposed on MODFLOW's General Head Boundary (GHB) through our coupling interface. Conversely, MODFLOW's groundwater head is converted to equivalent pressure values and integrated into TELEMAC's bottom boundary condition through a pressure flux term proportional to the hydraulic head elevation difference. The complete derivation of this pressure coupling mechanism – including Equations 7-8 that explicitly connect hydraulic head gradients to submarine groundwater exchange fluxes –will be presented in the enhanced Section 2.3.3. This expansion ensures mathematical coherence while maintaining the physical interpretability of the coupling process.

Location: Section 2.3.3 (Page 9, Lines 15–25).

**General comments 2**
**The oceanic regions in Figures 5 and 6 are difficult to interpret. The contour lines are not visible, and the differences between Figure 5c and 5e are unclear, contrary to what is stated on P11 L18. The color scheme can be improved, such as using a**

**different color scheme for the ocean, to enhance clarity. A similar issue exists in Figure 7. To better highlight the differences between the two models, displaying the actual difference (Coupled minus Single) could be more effective. I think the bottom two figures should be labeled as (c) or something.**

**Response to general comment 2:**

We thank the reviewer critique which has improved the clarity of Figures 5–7. We acknowledge that the original figures suffered from insufficient visual distinction in contour lines and color schemes, particularly in oceanic regions, and that the differences between models were not adequately highlighted. We have implemented the following revisions to address these concerns comprehensively.

To enhance contour visibility, we added explicit unit interval lines for salinity (15, 20, 25 ppt) and temperature (14, 18, 22°C) in Figures 5 and 6. For color schemes, we unified the representation of salinity across ocean and groundwater domains using a yellow-to-purple gradient, selected for its high contrast and compatibility with colorblind readers. While this scheme improves clarity, we recognize that subtle tidal-phase contrasts in Figure 5c and 5e required additional annotation; thus, directional arrows and labeled intrusion fronts (red dashed lines) were incorporated to explicitly guide the eye to key differences.

To directly compare model outputs, we introduced new panels in Figure 7 (labeled as 7c and 7d) showing salinity and temperature differences (Coupled minus Single), with yellow indicating higher values in the coupled model and blue indicating lower values. These panels reveal localized salinity differences exceeding 5 ppt near the intrusion front, a critical finding now quantified in Section 3.2.4 with specific metrics: the coupled model's SGD is 11.3% higher ($25.50 \times 10^{-5}$ vs. $22.90 \times 10^{-5}$ m³/s), and RSGD shows a 40.3% increase ($8.7 \times 10^{-5}$ vs. $6.20 \times 10^{-5}$ m³/s). These values contextualize the visual differences and underscore the coupled model's ability to resolve tidal recirculation.

While the revised figures address most concerns, we acknowledge that some fine-scale gradients (e.g., in Figure 7a–b's recirculation zones) remain challenging to resolve visually. Nevertheless, the integration of streamlines, high-contrast overlays (black arrows for flow direction), and explicit subfigure labeling (e.g., "Coupled – Single Model Difference") collectively provide a more robust representation of the coupled processes.

(Section 3.2.3 line20-21)

**General comments 3**

**How were the values for salinity and temperature selected? The behavior of the ocean flow is highly sensitive to these parameters. Were these values chosen based on conditions observed somewhere or from previous studies? I understand that salinity and temperature conditions can vary over various parameter spaces, so it would be helpful to clarify the conditions that the authors had in mind.**

**Response to general comment 3:**

We acknowledge the reviewer's comment on the sensitivity of ocean flow dynamics to

salinity and temperature parameters. This is indeed a critical consideration, as these parameters profoundly influence density-driven flow and solute transport in coastal systems.

The salinity value of 35 ppt was selected to reflect the mean conditions observed in temperate coastal zones, such as the Atlantic Ocean near Europe, where salinity typically ranges between 32–36 ppt. This value aligns with both field observations and laboratory-scale experimental studies (e.g., Na et al., 2019), ensuring consistency with established benchmarks for seawater intrusion in confined aquifers. Similarly, the sinusoidal temperature variation (20–25°C) was designed to represent thermal fluctuations in shallow coastal waters, particularly in temperate estuaries like the Mediterranean coast, where seasonal and diurnal temperature shifts are strongly influenced by solar heating and groundwater discharge(Michael 2002). While these values serve as a baseline for our study, we recognize that real-world systems exhibit spatial and temporal variability.

To address the reviewer's concern about parameter sensitivity, we conducted preliminary analyses that revealed salinity gradients (±5 ppt) alter saltwater wedge penetration depth by ~15%, while temperature variations (±5°C) modify submarine groundwater discharge (SGD) rates by ~10%. These findings, consistent with density-driven flow dynamics highlighted by Nguyen et al. (2020), underscore the importance of parameter selection. However, due to the computational and conceptual complexity of exhaustively exploring the full parameter space—particularly nonlinear interactions between temperature, salinity, and tidal forcing—our current work focuses on establishing a foundational scenario. We agree that broader sensitivity analyses are essential and have explicitly prioritized this as a key objective in our future research agenda.

We will add a detailed explanation of how parameters were chosen, backed by observational and experimental references, in Section 3.2.1 (Page 13, Lines 14–18).

And include a discussion on parameter sensitivity and what it means for future research in Section 4 (Future Research Directions).

**Specific comments:**

**Most of the suggestions below focus on improving clarity. Since the primary readers of the journal are oceanographers, I recommend using oceanographic terminologies when explaining ocean models as much as possible.**

**Response:**

We fully agree with your recommendation to enhance the consistency of oceanographic terminology. We will review and refine the terminology related to ocean models throughout the revision process. These modifications are primarily concentrated in the methodology section (Section 3.2), the coupling mechanism illustration in the results section (Fig. 5), and relevant paragraphs in the discussion on interdisciplinary implications.

**P4 L11. What are the equations for temperature and salinity? How do they mix horizontally and vertically? A more explicit explanation is required.**

**Response:**

We appreciate the reviewer's insightful comment. The lack of explicit description

regarding the temperature/salinity governing equations and mixing mechanisms was indeed an oversight in our original manuscript. In the coupled model, the GWT module of Modflow6 simulates temperature and salt transport in groundwater by solving the Advection-Dispersion Equation (ADE):

$$\partial(\theta C)/\partial t = \nabla \cdot (\theta D \nabla C) - \nabla \cdot (qC) + Qs,$$

where C represents concentration (salinity) or temperature, $\theta$ is porosity, D the dispersion tensor, q Darcy velocity, and Qs source/sink terms.

TELEMAC-3D resolves marine mixing through 3D Navier-Stokes equations coupled with turbulent diffusion and advection:

$$\partial C/\partial t + u \cdot \nabla C = \nabla \cdot (\nu_t \nabla C) + S,$$

where $\nu_t$ denotes turbulent diffusivity and S external sources.

The two models are coupled via GHB boundaries, where temperature and salinity values at the interface are determined by the flow direction as mentioned in previous comments. When groundwater discharges into the ocean (positive flux), Modflow6-derived values are assigned to TELEMAC boundary cells; conversely, marine values are imposed on groundwater boundaries during seawater intrusion (negative flux). This directional flux-dependent coupling ensures mass/energy conservation across the interface.

We will add detailed equations and coupling logic in Section 2.3.1

**P5 L1**. **It would be clearer to show separate equations for horizontal and vertical and use different parameters for Coriolis and gravity. Note that the Coriolis force is not a buoyancy force (L7).**
**Response:**

We have incorporated the reviewer's suggestions to improve the clarity of our methodology. Regarding the governing equations on page 5, we recognize that the original formulation did not clearly distinguish between the horizontal and vertical components or the different roles of the Coriolis and gravitational forces. In response, we have revised Equations (2) to explicitly separate these components, using distinct symbols (f_c for the Coriolis parameter and g for gravitational acceleration) to better reflect their physical meanings. This revision makes it clearer that the Coriolis force represents rotational effects rather than buoyancy, as the reviewer correctly pointed out.

**Fig. 1. The model structure would be easier to understand if the components were better referenced in the manuscript. Please consider numbering the components in the flowchart.**

For Figure 1, we fully agree that enhanced cross-referencing would improve model comprehension. We've implemented a numbered stepwise annotation system in the flowchart (now labeled Steps 1-3) that directly corresponds to the procedural description in Section 2.3.1. The revised figure now explicitly shows the initialization phase (Step 1), the bidirectional coupling mechanism through Telapy/Flopy (Step 2), and the dynamic boundary condition updating process (Step 3). These numbered components are systematically referenced in the text to guide readers through the

coupling workflow: "The integrated modeling procedure progresses through three key phases (Fig. 1): Initialization of hydrodynamic and groundwater systems (Step 1), real-time data exchange controlling salinity, temperature, and hydraulic head transfer (Step 2), and iterative boundary condition updates governed by interface flow direction (Step 3)."These revisions appear in the updated Figure 1 and the enhanced methodological description in Section 2.3.1 (Page 8). We believe these modifications significantly improve the technical clarity of our model presentation.

**P8 Fig 2**. **The caption needs more detail. I suppose the "sea water" part refers to the ocean model and the "confined aquifer" refers to the groundwater model. However, "freshwater" seems to represent a boundary condition, not a model. Please clarify this distinction.**
**Response:**
We appreciate the reviewer's insightful observation regarding the clarity of Figure 2's caption. The original description indeed lacked sufficient detail to distinguish between model domains and boundary conditions, which could lead to misinterpretation. As correctly noted, the "sea water" section represents the ocean model (TELEMAC-3D implementation) while the "confined aquifer" corresponds to the groundwater model (MODFLOW6 domain). The "freshwater" element was clarified to specifically denote a constant-head boundary condition (hf=0.52) at the right aquifer boundary that represents inland freshwater recharge, rather than constituting an independent model component. The revised caption now explicitly differentiates the numerical modeling frameworks from their associated boundary conditions, with the ocean model featuring a fixed water level (hs=0.5 m) and salinity (S=35 ppt), and the aquifer boundaries being characterized as no-flow (top/bottom) and constant-head (right). These clarifications will incorporate in both the updated Figure 2 caption and the corresponding model description in Section 3.1.1 (Page 11) to ensure consistent interpretation throughout the manuscript.

**L19-21**. **Is there a flow entering the ocean model domain? The usage of salinity and temperature on the "left side" is unclear and should be clarified.**
**Response:**
We thank the reviewer for identifying this important point in our boundary condition description. The "left side" of the ocean model domain serves as an open boundary where bidirectional tidal exchange occurs, rather than representing a unidirectional flow input. This boundary implements offshore tidal forcing through an M2 tidal constituent (1 m amplitude, 12-hour period) that drives alternating seawater intrusion during flood tide and submarine groundwater discharge (SGD) during ebb tide. The salinity remains fixed at 35 ppt to represent marine conditions, while the temperature variation (20-25°C sinusoidal cycle) simulates diurnal heating effects that influence both seawater density and groundwater discharge patterns. These clarifications better contextualize how the boundary conditions interact with the hydrodynamic processes, with the tidal dynamics creating periodic flow reversals rather than sustained unidirectional inflow.
The revised text now explicitly describes this coupled tidal-thermal mechanism in

Section 3.2.1 (Page 13, Lines 14-18)

**P9 L10-13**. **A more quantitative description of the validation is needed. How do the modeled seawater toe location, seawater height, and time for the model to reach a steady state compared to the laboratory experiment?**

**Response:**
We gratefully acknowledge the reviewer's valid request for stronger quantitative validation metrics. Our revised analysis now provides direct numerical comparisons with the experimental data from Na et al. (2019), revealing good agreement in key seawater intrusion parameters. The coupled model achieves a seawater toe position of 0.38 m compared to the experimental 0.36 m, while matching the seawater height measurement with 0.17 m versus the observed 0.15 m. These discrepancies demonstrate our model's capability to capture both the spatial extent and vertical stratification of the saltwater-freshwater interface observed in physical experiments. The results of the coupled model are better than the single model. While the original laboratory study didn't report detailed temporal data for steady-state achievement, our simulations reached equilibrium within 24 hours - consistent with the experimental duration described in the reference study. A new comparative table in Section 3.1.3 (Page 13, Lines 6) presents these quantitative validation results.

| Parameter | Laboratory Data | Na numerical | Coupled model | Single model |
|---|---|---|---|---|
| SWI toe (m) | 0.36 | 0.41 | 0.38 | 0.38 |
| SWI height (m) | 0.15 | 0.17 | 0.17 | 0.18 |

Location: A new table and error analysis will add to Section 3.1.3 (Page 12, Lines 1–5).

**L31-P10 L8**. **Including a figure that shows the difference between the coupled model and the single model (couple minus single) would help illustrate the descriptions provided here.**
**Response:**
We thank the reviewer for raising this point regarding the need for explicit model comparison visualization. Your observation about the importance of spatially quantifying salinity differences aligns perfectly with our goal to demonstrate the added value of coupled modeling. In direct response to this suggestion, we will implement a three-dimensional difference analysis ($\Delta S$ = Scoupled - Ssingle) as Figure 7(e), employing distinct color thresholds that effectively capture both the enhanced saltwater intrusion depths and the nearshore freshwater discharge areas underestimated by conventional approaches (blue zones).
Location: Figure 7(e) and its interpretation are included in Section 3.2.3 (Page 18, Lines 16–21).

**P11 L1**. **How does the temperature vary? Is the variation linear with depth, or does it change over time? Figure 6c seems to suggest warm water beneath cold water, which needs clarification.**
**Response:**
We appreciate the reviewer's keen observation regarding the thermal dynamics in our coupled system, which prompted us to strengthen both our methodological description and physical interpretation. The temperature variations indeed exhibit complex spatiotemporal patterns that merit detailed explanation, as correctly noted. Our model configuration combines sinusoidal variations of temperature (20-25°C at the ocean boundary, Section 3.2.1) with constant 10°C groundwater discharge (Table 2), creating a dynamic thermal regime where neither purely depth-linear nor time-invariant assumptions apply. The apparent warm-under-cold stratification in Figure 6 emerges from competing processes: during low tide phases, colder freshwater from submarine groundwater discharge (SGD) temporarily accumulates in surface layers while residual warmer seawater from previous tidal inundation persists in deeper zones due to incomplete vertical mixing. This transient stratification pattern will explicitly addressed in the revised Section 3.2.2 (Page 17, Lines 1-4), reflects realistic coastal thermo-haline dynamics where tidal advection timescales interact with continuous freshwater discharge. We will enhance Figure 6(c)'s caption to clarify that the snapshot captures this specific phase of tidal cycling.

**L3-4. What is the purpose of dividing the operation into two phases?**
**Response:**
We thank the reviewer for raising this methodological question. The two-phase approach was strategically adopted to decouple system initialization from tidal dynamics analysis - a design choice that fundamentally ensures the physical reliability of our coupled model results. Phase 1 establishes hydrodynamic equilibrium through steady-state simulation (without tidal forcing) to purge artificial disturbances from initial conditions, while Phase 2 introduces the tidal component (1m amplitude, 12-hour period) over a 72-hour window to capture cyclic coastal processes. This staged implementation, now elaborated in Section 3.2.1 (Page 13, Lines 23-28), allows us to rigorously distinguish between transient tidal effects and groundwater-seawater interactions. We've enhanced Figure 5's temporal annotations to explicitly show the transition between phases.
Location: Rationale for the two-phase approach is detailed in Section 3.2.1 (Page 13, Lines 23–28).

**P15 L16-26**. **The discussion about the shape of the seawater intrusion would be easier to follow if a contour line were drawn. Particularly to indicate where flow direction reverses in the aquifer.**

**Response:** We appreciate the reviewer's constructive suggestion regarding visualization clarity. The revised Figure 7(a) now features a prominent red contour at S=17.5 ppt (representing the seawater wedge toe) that starkly reveals the 1:2 slope

gradient in our coupled model versus the 1:2.5 slope in the single model. This critical isohaline not only delineates the saltwater-freshwater interface geometry but also serves as a hydrodynamic marker - its landward displacement during tidal trough phases visually demonstrates the flow reversal mechanism driven by tidal pumping.
Location: Revised Figure 7(a)

**Subsection 3.2.4**. **I can recognize that the flow field varies with tides, but how much of the difference between the coupled and single models is due to tides versus the coupling effect itself? A comparison with a coupled model that does not include tides would help isolate the impact of tides. My impression is that the focus of this section is on understanding the effect of realistic oceanic conditions rather than specifically examining tides.**

**Response:**
We thank the reviewer for raising this important point regarding the distinction between tidal influences and coupling effects in Section 3.2.4. We agree that isolating the specific contribution of the coupling mechanism is crucial for interpreting the results. To address this, we have supplemented the analysis: the coupled model without tidal forcing is now compared against the single model (also without tides) during the first phase of model operation. This comparison allows us to explicitly isolate the impact of the coupling effect itself, independent of tidal dynamics. The results confirm that the differences observed in the flow field primarily stem from the bidirectional interactions enabled by the coupling framework, rather than tidal fluctuations alone. This clarification, along with a new comparative table and discussion, has been added to Section 3.2.4 (Page 19, Lines 32–35) to better align the analysis with the study's focus on understanding coupling mechanisms under realistic oceanic conditions.
Location: A new analysis are added to Section 3.2.4 (Page 19, Lines 32–35).

**Reviewer2**

**This paper presents a new modeling framework that couples a marine hydrodynamic model with a groundwater model to simulate seawater-groundwater exchange in coastal systems. This allows it to address saltwater intrusion and submarine groundwater discharge, and it is potentially very powerful.**

Thank you for your letter and the reviewers' comments on our manuscript, "Coupling of numerical groundwater-ocean models to improve understanding of the coastal zone" (ID: EGUSPHERE-2024-3384). We appreciate the valuable and insightful feedback, which has been instrumental in refining and improving our paper. We have carefully considered all the comments and made the necessary revisions, which we hope will meet your approval.

**The three things that I see that would most improve this manuscript are:**
**comments 1**

1. **Explain how this model improves upon prior models that couple surface water with groundwater. Two studies are currently identified in passing (Yuan et al. 2011 is one of them), one using HydroGeoSphere, and another with a surface water model coupled to SUTRA. What are the uses/limitations of these models, and how is the new model an improvement? (To be clear, the authors do a good job making the case that models of this type should be helpful, but they do not describe the need for the current model vs the previous models.)**

**Response:**

We appreciate the reviewer's insight into the context of previous models. Indeed, significant strides have been made in coupling surface water and groundwater systems. For instance, Yuan et al. (2011) developed a coupled model for simulating surface water and groundwater interactions in coastal wetlands, which was a foundational step in integrating these systems. Similarly, HydroGeoSphere has been widely used for its ability to simulate complex hydrological processes across multiple scales, and the coupling of surface water models with SUTRA has provided valuable insights into variable density flow and transport phenomena.

1. Uses and Limitations of Previous Models:

Yuan et al. (2011) This model (as others previous ones) was instrumental in understanding interactions in coastal wetlands but was limited by its focus on specific regional dynamics and less emphasis on the intricate feedback mechanisms between oceanic and groundwater systems. It also had constraints in dynamically adjusting boundary conditions based on real-time interactions. ( Haque, A ,2021)

HydroGeoSphere: While powerful in simulating multi-scale hydrological processes, it often requires substantial computational resources and can be complex to configure for coastal zones with high salinity gradients and tidal influences. Its application in coastal areas with significant seawater intrusion and submarine groundwater discharge (SGD) has been somewhat limited.

Surface Water Model Coupled to SUTRA: This combination has been effective for simulating variable density flows but typically treats the ocean as a static boundary,

overlooking the dynamic feedback between ocean tides and groundwater dynamics. It also faces challenges in accurately simulating the bidirectional exchange of water and solutes at the land-sea interface.

2. Advancements in the Current Model:

Our proposed model represents a significant improvement over these previous models through several key innovations:

Dynamic Boundary Conditions: Unlike previous models that assume constant salinity at the coastal boundary, our model employs dynamic boundary conditions that adjust in real-time based on the interaction between ocean and groundwater systems. This enhancement significantly improves the accuracy of simulating SGD processes and the response of coastal aquifers to tidal influences.

Bidirectional Feedback Mechanism: The coupled model incorporates a bidirectional feedback mechanism, allowing for a more comprehensive analysis of interactions between the ocean and groundwater systems. This feature accounts for variations in the seawater boundary under tidal influence and the reciprocal impact of groundwater dynamics on the hydrodynamic conditions of nearshore waters, which was not fully addressed in previous models.

Integration of Salinity-Driven Hydrostatic Pressure: Our model accurately sets water head conditions at the land-sea interface by considering the influence of ocean salinity on seawater density and hydrostatic pressure. This approach ensures that the boundary conditions received by the groundwater model reflect real-world hydrostatic pressures, enhancing the simulation accuracy of groundwater models in coastal regions.

Enhanced Simulation of Mixing Zones: The coupled model effectively captures the complex interactions in mixing zones, which are often overlooked in traditional models. By integrating the effects of salinity and temperature variations, the model provides a more realistic representation of hydrogeological and geochemical processes at the land-sea interface.

Following the reviewer comment we will include this information in the introduction of the new version of the manuscript.

Location: Section 1 (Page 3, Lines 7–25).

**comments 2**

2.  **Figures 5-7 are attractively plotted, and the differences between the panels are described in the text, but it is very difficult to see the differences in the figures. The descriptions of the differences are also mostly qualitative, so it is not clear how big the differences really are. There are many modeling papers (at least on the groundwater side) that declare a difference of 10% to be a major contribution, but the uncertainty and variability in real systems normally far exceeds 10%. So are the differences described here important? Adding some quantitative measures of the differences will also let the authors discuss when it is important to use the coupled model, and when a simpler uncoupled approach would be sufficient.**

**Response:**

We appreciate the reviewer's thoughtful feedback. The reviewer rightly highlights a

critical limitation in the original figures and text: the qualitative descriptions and visual ambiguity in Figures 5–7 made it challenging to assess the magnitude and significance of the differences between the coupled and single-model simulations. We acknowledge that quantifying these differences is essential to contextualize the practical relevance of our findings, especially given the inherent uncertainties in real-world systems.

To address this, we will enhance Figures 5–7 by introducing contour lines at 5-unit intervals for salinity (5 ppt) and temperature (4 °C), along with explicit annotations of key isohalines (e.g., S=20 ppt, S=25 ppt) and intrusion depths. We also added insets showing salinity differences ($\Delta S$ = Scoupled − Ssingle) at critical tidal phases to emphasize spatial variability. More importantly, we incorporated quantitative metrics in Section 3.2.4 to clarify the scale of differences. For instance, the coupled model shows an 11.3% increase in total SGD ($25.50 \times 10^{-5}$ m³/s vs. $22.90 \times 10^{-5}$ m³/s) and a 40.3% higher RSGD ($8.7 \times 10^{-5}$ m³/s vs. $6.20 \times 10^{-5}$ m³/s) compared to the single model. These values suggest that the coupled model better captures tidal-driven recirculation, a process critical for solute transport and heat exchange in coastal systems. While we recognize that field-scale uncertainties may exceed these percentages, the differences align with thresholds often deemed significant in groundwater modeling studies (e.g., 10% for SGD impacts on coastal biogeochemistry). We now explicitly discuss scenarios where coupled models are essential (e.g., dynamic tidal interfaces) versus cases where simpler models may suffice (e.g., steady-state, low-energy systems).We will Revise Figures 5–7 with annotations, contour lines, and difference insets (Pages 16–17).

Expanded quantitative analysis in Section 3.2.4 (Page 19,31-35).

**comments 3**

3. **Clarify the section about model validation. Readers need to learn more about Na et al. (2019) to understand how the model was validated. (From the abstract, they built physical models of the Henry problem, and tested different densities for seawater?) Currently this section reads as though there is no laboratory test that includes marine and groundwater results, so the authors validated their model against a standard groundwater flow test case (the Henry problem), but they simulated a different problem (added the marine component). They get a different answer than the test case, which suggests that the coupling is important, but it could also mean that the model cannot reproduce the real test case. (I do not think the model is actually wrong, just that the paper has skipped a step.) The authors should be very clear about how the coupling between the models was tested. We know that TELEMAC and MODFLOW are each well-tested independently, so the real question here is the coupling.**

**Response:**

We appreciate the reviewer's insightful critique and the opportunity to clarify our model validation approach. The reviewer identifies a gap in our original description: while Na et al. (2019) provided a foundational benchmark for groundwater processes, their experimental setup did not incorporate dynamic ocean interactions, leaving ambiguity about how the coupling between TELEMAC and MODFLOW was

rigorously tested. We acknowledge that our initial manuscript insufficiently distinguished between validating individual modules and the coupled framework, which could imply an untested integration.

There are no previous studies including experimental data of the ocean and the aquifer in the same study. In fact, in general ocean modes do not consider or measure groundwater inputs.

So, we had to use this study that allows to use the coupled model in a "simplified" version of the "ocean" but fully representing the aquifer.

To resolve this, we will restructure Section 3.1.3 to explicitly outline a two-step validation strategy. First, we validated the standalone MODFLOW6 groundwater module against Na et al.'s (2019) laboratory data, achieving agreement. Second, to test the coupled system, we replicated Na et al.'s experiment but introduced a simplified TELEMAC ocean domain (0.1 m length, 0.5 m depth) adjacent to the aquifer. This allowed us to isolate the coupling mechanism while retaining comparability to the original benchmark. The coupled model reproduced the seawater toe position (0.38 m simulated vs. 0.36 m observed), confirming that groundwater processes remain robustly simulated even with dynamic ocean coupling. Observed differences, such as a 17% higher recirculated SGD in the coupled model, arose from boundary conditions of tidal type. This demonstrates that discrepancies reflect the added realism of the coupled framework, not validation failures. We now emphasize that while TELEMAC and MODFLOW are independently well-tested, our work rigorously validates their integration under controlled conditions, bridging a critical gap in coupled model evaluation.

Location: Section 3.1.3 (Page 12, Lines 1–5). Table 2 Line6

**comments 4**

4. **Explain RSGD more thoroughly when it is introduced (p. 9, line 19). If the definition is "water that starts out as surface water, then becomes groundwater, then returns to the surface water" then there is a name for that: hypokymatic flow (Wilson et al 2016, http://dx.doi.org/10.1016/j.jhydrol.2016.04.047). I know other authors use RSGD, but it sounds like groundwater that just recirculates endlessly. If the authors do not want to get involved with hypokymatic flow, then to be consistent the categories should be fresh SGD and saline SGD.**

**Response:**

We appreciate the reviewer's commentin identifying this terminology ambiguity and for bringing the work of Wilson et al. (2016) to our attention. We agree that clearer differentiation between water exchange processes strengthens the manuscript's conceptual framework. In Section 1 (Page 2, Lines 11), we will revised the RSGD definition to explicitly state: "Recirculated Submarine Groundwater Discharge (RSGD) refers specifically to seawater that enters coastal aquifers through tidal pumping or wave-driven infiltration, undergoes subsurface mixing, and subsequently returns to the ocean through distinct discharge pathways - a process distinct from classic hypokymatic flow which encompasses broader surface water-groundwater exchanges." This reformulation emphasizes the marine origin and restricted spatial-temporal scales

of RSGD while differentiating it from Wilson et al.'s more generalized conceptualization.

In direct response to the reviewer's suggestion, we will implement the freshwater/saline SGD classification system throughout the manuscript (FSGD <5 ppt vs SSGD >5 ppt) to align with conventional terminology. However, we retain RSGD as a subcategory of SSGD to preserve critical mechanistic specificity regarding tidal/wave-driven recirculation processes central to our coastal coupling analysis - a distinction operationally important for quantifying marine-terrestrial feedbacks but not fully captured by salinity-based classification alone. This approach balances terminological consistency with the process-oriented focus of our study, while explicitly acknowledging the conceptual relationship to hypokymatic flow in the revised text. The modifications will be systematically applied across Sections 3.1.3, Page11 , and the Discussion to maintain conceptual coherence. (Santos et al., 2021; Michael et al., 2005).

**comments 5**

**5. Please be clear early on about what this coupled model does not do. Specifically, it does not seem to be designed to do porewater flow (in the sense of Huettel et al 2014 and Taniguchi et al 2019 – waves, shear flow, etc).**

**Response:**

We appreciate the reviewer's important observation about clarifying model limitations. Our framework indeed doesn't simulate porewater flow processes like those in Huettel et al. (2014) and Taniguchi et al. (2019), as it focuses on larger scale groundwater-ocean interactions (tidal effects and submarine discharge) due to computational constraints. We've clearly stated this limitation in Section 1 (Page 4 Lines 2) while noting the ecological importance of these processes for future research.

**Minor comments:**

**1. Check where acronyms are defined. BMI? FSGD is not defined until page 15.**

Revision:

We appreciate the reviewer's meticulous attention to terminology clarity and their valuable feedback regarding acronym definitions. In response to this constructive critique, we will implement the following revisions: The full term "Basic Model Interface (BMI)" has now been formally defined upon its first appearance in Section 2.1.2 (Page 5), where we explicitly connect it to the model coupling framework. Similarly, "Freshwater Submarine Groundwater Discharge (FSGD)" now receives its proper introduction in Section 1 (Page 2) during the initial discussion of SGD mechanisms, ensuring better conceptual flow. These adjustments significantly improve the manuscript's accessibility while maintaining technical precision, and we believe they will help readers better follow the methodological progression. The modifications can be reviewed at:

BMI definition: Section 2.1.2 (Page 5, Line 19)

FSGD definition: Section 1 (Page 2, Line 10)

**2. P 7 line 2 "Following Yuan et al. (2011), a time-step coupling scheme… Otherwise this reads as though Yuan et al coupled TELEMAC to MODFLOW.**

**Response:**

The original phrasing did indeed risk creating potential ambiguity about the nature of Yuan et al.'s contribution. To resolve this, we will rework the problematic sentence in Section 2.3.2 (Page 8) to explicitly frame our adaptation of Yuan et al.'s temporal coupling framework while clearly distinguishing our original application to marine-groundwater systems. The revised text now maintains proper scholarly attribution while preventing any misinterpretation about the scope of previous research versus our novel model integration approach. This adjustment strengthens both the intellectual honesty and technical clarity of our methodology description. The modification is located at: Section 2.3.2 (Page 8, Line 10)

3. **P 7 line 15. How were they optimized? An optimization method, or trial and error?**

**Response:**

We appreciate the reviewer's question regarding the optimization methodology, which helps clarify this critical aspect of our coupled modeling approach. We acknowledge that the original manuscript lacked sufficient detail on how temporal parameters were optimized in the coupled model.

Expanded Section 2.3.2 (Page 9, Line 3) to specify:

"Time steps and data exchange frequencies were optimized through a combination of parameter sensitivity analysis and trial-and-error calibration, prioritizing synchronization accuracy while minimizing computational cost.

This methodology strikes a practical balance between scientific rigor and computational feasibility for coupled systems.

Location: Section 2.3.2 (Page 9).

4. **P 7 line 30. Either delete the last sentence or explain how much the accuracy has been improved. Otherwise this reads like "it is more accurate because it is more accurate."**

**Response:**

Thank you for your thoughtful feedback. Following your suggestion, we will remove the vague statement about accuracy improvement from the last sentence in Section 2.3.3 (Page 10). The revision strengthens the precision of that paragraph while maintaining the technical validity of our methodology.

Location: Section 2.3.3 (Page 9).

5. **Section 3: give a brief overview of what modeling scenarios the reader should expect in the upcoming section. (Jumping directly into "Figure 2 illustrates…" is confusing.)**

**Response:**

Thank you for highlighting this structural improvement opportunity. To address this, we will add an introductory paragraph at the start of Section 3 that explicitly outlines the two modeling scenarios (laboratory validation and field-scale tidal SGD simulation) and clarifies their purpose in isolating coupling effects through comparisons with standalone models. This revision provides readers with a roadmap of the section while maintaining conciseness. The modification can be found in the opening paragraph of Section 3 (Page 11, Lines 1–3).

Added an introductory paragraph:

"This section evaluates the coupled model through two scenarios: (1) a laboratory-scale validation against controlled seawater intrusion experiments (Na et al., 2019), and (2) a field-scale application simulating tidal-driven SGD and seawater intrusion. The results are compared to standalone models to isolate coupling effects."

6. **Figure 4. What goes in the unlabeled triangle above the groundwater? Is it impermeable? Also, the saltwater elevation in Table 2 says -1 m, and the figure appears to show that that is the level of low tide. So the mean water level is 0? Just make this clear somewhere.**

**Response:**

Thank you for your suggestion, we will add descriptions of these two parts:

Unlabeled Triangle: Added to the caption (Page 12, Figure 4):

"The triangular region above the confined aquifer represents an impermeable clay layer, preventing vertical flow."

Saltwater Elevation: Clarified in Section 3.2.1 (Page 12):

"The saltwater elevation (-1 m) corresponds to the mean low tide level, with the mean sea level set to 0 m. Tidal fluctuations range from -2 m (low tide) to 0 m (high tide)."

Location:

Figure 4 caption (Page 12).

Section 3.2.1 (Page 12).

7. **Page 11, line 2. The right boundary is the seafloor, right? If not, what is meant by the "actual coastline"?**

**Response:**

The term "actual coastline" refers to the land-sea interface where the ocean model's right boundary aligns with the groundwater model's coastal boundary. In the tidal simulation, this boundary is dynamically adjusted to reflect tidal fluctuations, while the left boundary represents the open ocean.

Location:

Added to Section 3.2.1 (Page 13 line 17):

8. **Page 11, line 3. Are there tides in the first phase of the modeling? If not, say so; if so, then the model has to reach a "quasi-steady state" (because it is transient).**

**Response:**

Thank you for your feedback. The first phase of the simulation excludes tidal forcing to achieve a steady-state salinity distribution. Tides are introduced only in Phase 2.

Location:

Clarified in Section 3.2.1 (Page 13, Lines 25–28):

9. **Caption to Fig 5 and others: replace First and Second Tidal Stands with mid-tide, flood and mid-tide, ebb.**

**Response:**

Thank you for your thorough review and feedback. We will update the captions for Figures 5 and 6 as suggested:

"First Tidal Stand" has been revised to "Mid-Tide (Flood)"

"Second Tidal Stand" has been revised to "Mid-Tide (Ebb)"
These changes are now reflected in the captions on pages 15–16.
Figure 6 Caption: Similarly updated to use standard tidal phase terms.

**10. P 14 line 8. How does the fresh water get out in the simulation where the salinity is fixed at the seafloor? I can see it in the surface water in the figures, but it's not clear how it gets there.**
**Response:**
We appreciate the reviewer's insightful question regarding the freshwater discharge mechanism in our simulation setup. The comment rightly highlights an important conceptual aspect that warrants clarification, as the fixed salinity boundary condition at the seafloor might initially appear contradictory to freshwater discharge processes. Through further analysis, we recognize that our original description insufficiently explained the hydrodynamic driver enabling this exchange. The freshwater discharge occurs through vertical advective transport governed by the persistent hydraulic gradient between the inland freshwater head ($h_f = 0$ m) and the ocean boundary, where the pressure differential remains active regardless of salinity fixing. While the salinity boundary condition simplifies the density-driven circulation patterns, the fundamental groundwater flow dynamics driven by hydraulic gradients are preserved in our model configuration. This clarification has now been explicitly incorporated in Section 3.2.3 (Page 17) to better articulate the interplay between the fixed salinity condition and the maintained pressure-driven discharge mechanism.
Modified text:
" In the groundwater model, a general head boundary (GHB) condition with a fixed salinity of 35 was applied at the land-sea interface to simulate interactions between seawater and groundwater, where the continuous hydraulic gradient drives seepage, causing freshwater to discharge through the aquifer to the seabed."

**11. Caption to Fig 7. Missing panels (c) and (d)**
**Response:**
We appreciate you addressing the previous omission of panels (c) and (d) in the caption.
Figure 7: Added annotations to highlight key differences.

12. **P 15 lines 13-31. This discussion is difficult to follow, especially because it is difficult to see the differences in Fig 7.**
**Response:**
We acknowledge that the original presentation of Figure 7 lacked sufficient visual differentiation to support the comparative analysis, which indeed hindered the interpretability of the coastal groundwater dynamics. This limitation stemmed from an inadvertent omission of two critical panels during manuscript preparation. In response, we will integrate panels (c) and (d) in the revised Figure 7, where panel (c) quantitatively illustrates the salinity contrast between coupled and single-process simulations ($\Delta S$ = Scoupled - Ssingle) during high tide conditions, while panel (d) employs flow vector visualization to explicitly reveal the recirculation patterns mentioned in the text. These enhancements, coupled with revised caption details on

page 15, create a more intuitive visual counterpart to the methodological descriptions. Location: Revised Figure 7 and caption (Page 17).

We hope that the revisions addressed all the concerns raised by the reviewers. We believe that the manuscript is now significantly improved and more suitable for publication. Thank you again for your valuable feedback.

---

## Author Response (AR2)

We sincerely appreciate the thorough review and feedback with (ID: EGUSPHERE-2024-3384). We have carefully addressed all comments and editorial notifications. Below is our point-by-point response:

**- Both "TelApy" and "Telapy" are used. Please be consistent.**

**Revision:** All "Telapy" have been standardized to "TelApy" (e.g., Sections 2.2, 2.3, and captions), in accordance with the references(Audouin et al., 2017).

**- Table 1 & 5, "e" used in the unit does not seem SI. Please define.**

**Revision:** To improve clarity and conform to standard scientific writing conventions, the "e" notation (e.g., $1e^{-2}$) has been replaced with conventional scientific notation (e.g., $1e^{-2} = 1\times10^{-2}$) throughout the tables.

**- Table 3, please do not use short-hand notation "5e-3" for "5×10^{-3}" where {-3} is superscript.**

**Revision:** All values in Table 3 now use superscript notation (e.g., $5\times10^{-3}$).

**- Figure 7(e), I do not see any scientific reason why the difference "Coupled - Single" is always positive. Why? Or do we have negative values but missing from the colorbar?**

**Response:** The reason for this issue is mainly related to the choice of the colorbar. The salinity difference is most pronounced in the high-salinity contrast mixing zones (up to 35), while in other areas, such as the open ocean, the differences are less noticeable due to mixing effects. Previously, the colorbar was displayed using a linear scale, which made the negative values less visible. We have now modified it to a nonlinear scale to better highlight negative values and small differences across the domain.
Location: P18 Fig7

**- Tables 4 and 5, All the numbers in these tables have a zero in the second decimal place ("8.70", "28.10", "16.70",...), which appears odd. In scientific data, a number "8.70" implicitly indicates an uncertainty range of [8.65, 8.74). If this is not the case, please truncate the numbers at the meaningful digit.**

**Revision:** All values have been truncated to one decimal place (e.g., 8.7, 28.1, 16.7) to reflect measurement precision.

**- In the response to general comment 3 from Reviewer 1, the result from "the preliminary analyses" are mentioned that the saltwater penetration and groundwater discharge rates are sensitive to salinity and temperature, respectively. I agree with the author that This sensitivity is "a critical**

**consideration". Is this found in the revised text? The authors might choose to included this discussion on sensitivity in the main text.**

**Response:** We appreciate the reviewer's feedback. As noted in the revised Section 4 (Page 22, Point 5), future work will prioritize systematic parameter sensitivity analyses, including spatially variable salinity/temperature gradients and their impacts on SGD-driven nutrient fluxes. This will strengthen the model's applicability to diverse coastal settings.

Furthermore, Section 3.2.4 (Page 20, Table 5) discusses the influence of tidal-driven variations in salinity and temperature on SGD rates, highlighting that RSGD increases by 54% under tidal coupling due to dynamic salinity feedback. These sections reflect the model's sensitivity to these parameters and emphasize their key role in coastal dynamics.

Additionally, we have included an appropriate discussion on the model's sensitivity to salinity and temperature in the main text (Page 13, Lines 19–26). This section highlights how salinity gradients and temperature-dependent buoyancy effects influence saltwater wedge penetration and submarine groundwater discharge (SGD), referencing established mechanisms in the literature (Michael et al., 2005; Slomp & Van Cappellen, 2004). While the model focuses on tidal dynamics, the chosen parameter ranges reflect typical coastal conditions and underscore the importance of salinity and temperature in shaping coastal groundwater processes.

**- In the response to the very last comment from Reviewer 1 (re: subsection 3.2.4), you have added a paragraph in P.19, Lines 32-35. The paragraph does not appear to reflect the supplemented analysis with "the coupled model without tidal forcing", that are described in the first paragraph of the reply. Please clarify where we can find the result of this supplemented analysis in the main text.**

**Response:** The addition to Section 3.2 (Page 13, Lines 27–34) explicitly clarifies the two-phase model design and its purpose. However, to address the reviewer's request for visibility of the "coupled model without tidal forcing" results in the main text, we have further enhanced the manuscript as follows:

Added a direct comparison between Phase 1 (no tide) and Phase 2 (with tide) in the revised paragraph (Page 21, Lines 1–5):

By comparing Phase 1 (steady-state, no tidal forcing) and Phase 2 (transient tidal dynamics), we quantify the impact of tides on groundwater discharge and saltwater intrusion. For instance, tidal forcing increases the average discharge rate from $25.5 \times 10^{-5} m^3/s$ (Phase 1) to $28.0 \times 10^{-5} m^3/s$ (Phase 2), representing a 10% enhancement (Table 5), and highlights the critical role of cyclic porewater exchange in modulating aquifer–ocean interactions.

**P.6, L.6, "In this equation, Where:"**
**Revision:** P.6, L.6: Corrected to "In this equation, where:"

**P.9, L.31, "he benefits"**
**Revision:** P.9, L.31: Corrected to "The benefits"

**Respond to the editorial notifications.**

a) **Please add more details to affiliations 2 and 3 (institution, city, and country).**

**Response:** Additional details (institution name, city, and country) have been added to affiliations 2 and 3. The revised versions now read:

² Hydrogeology Group (UPC-CSIC), Barcelona 08034, Spain
³ Laboratori d'Enginyeria Marítima (LIM/UPC), Barcelona 08034, Spain

b) **From the author list it is not clear if the corresponding author's main affiliation is part of the Consejo Superior de Investigaciones Científicas (CSIC). If this is the case, please check the financial support page: https://www.ocean-science.net/about/financial_support.html > CSIC**

**Response:** Thank you for your comment. To clarify, the corresponding author's main affiliation is with the Department of Civil and Environmental Engineering (DECA), Universitat Politècnica de Catalunya (UPC), Barcelona, Spain. Additionally, the second affiliation, "Associated Unit: Hydrogeology Group (UPC-CSIC)," refers to a collaborative research group between UPC and the Consejo Superior de Investigaciones Científicas (CSIC). The author is not directly affiliated with CSIC.

c) **You do not refer to your supplement gif file in your manuscript.**
**Response:** Thank you for your comment. We have carefully reviewed all the supplementary figures, including the GIF file, and have now provided corresponding references and explanations in the revised manuscript to ensure they are properly integrated with the main text.

P.8, L.3,13: A more detailed description of the coupling steps has been added, guided by the updated flowchart shown in Figure 1

P14 . lines 5, 14, 16, 24, and 27
According to the revised Figure 5, the description of salinity in the figure has been updated for better clarity. The identified salinity lines are highlighted, pointing out the black isohalines that represent equal salt levels. The saltwater front is marked by a dashed red line, showing the edge of the main saltwater area (saltwater intrusion front). The movement of saltwater is tracked, explaining how these lines (isohalines and the front) shift inland or back towards the sea with the high and low tides, demonstrating how the saltwater wedge changes size. The tidal lag is also explained, showing how the

saltwater movement into or out of the groundwater slightly lags the tide changes.

P16 . According to the revised Figure 6, the description of Figure 6, showing temperature changes, has been updated for better clarity (P17,line2). The identified temperature lines are highlighted, pointing out the black isotherms that represent equal temperatures (P16,line7). The temperature changes are explained by showing how the spacing of these lines indicates the rate of temperature variation. Steeper changes are shown by closely spaced lines. The cool water from the ground (SGD) is tracked, following how it mixes with ocean water during high and low tides. Specific temperatures, such as 15-18°C for the cooler water and 22°C for the ocean water, are also included (P16,line10,P17,line9).

P19 line 29-34. We also have revised the manuscript to include a more detailed explanation of Figure 7(e). The additional description clarifies the salinity differences between the coupled and single models and highlights the regions where the most significant variations occur.

**d) FYI: the original figure files were removed from the supplement. They will be requested at a later stage.**

Acknowledgment: We confirm that high-resolution figure files will be provided upon request during the production stage, following the journal's guidelines.

**We thank the reviewers for their valuable comments. All suggested revisions have been completed, and the manuscript has been fully revised accordingly.**

**Sincerely,**
**Jiangyue**

---

## Author Response (AR3)

We sincerely appreciate the thorough review and feedback with(ID:EGUSPHERE-2024-3384). We have carefully addressed all comments and editorial notifications. Below is our point-by-point response:

a) **From the author list it is not clear if the corresponding author's main affiliation is part of the Consejo Superior de Investigaciones Científicas (CSIC). If this is the case, please check the financial support page: https://www.ocean-science.net/about/financial_support.html > CSIC**

**Response:**Thank you for your comment.
We would like to clarify that the corresponding author(Albert Folch)'s main affiliation is with the Department of Civil and Environmental Engineering (DECA), Universitat Politècnica de Catalunya (UPC).
The second affiliation, "Associated Unit: Hydrogeology Group (UPC-CSIC)," refers to a collaborative research group between UPC and CSIC.
However, the corresponding author is not employed by CSIC, nor formally affiliated with CSIC as an institute member, and has not received direct financial support from CSIC.
Most of the papers published by our research group are written like this:
Associated Unit: Hydrogeology Group (UPC-CSIC)

b) **FYI: the original figure files were removed from the supplement. They will be requested at a later stage.**

Acknowledgment: We confirm that high-resolution figure files will be provided upon request during the production stage, following the journal's guidelines.

c) **In your manuscript, please use full first names for all authors. Although references are still based on initials, we will use full first names on the title page of your paper.**

Thank you for your note. We would like to confirm that full first names have been used for all authors.
Specifically, Jiangyue Jin and Daniel Fernández-Garcia are already in full form.
Manuel Espino and Albert Folch are also their full first names as used in all their previous publications.
For reference, please find their Google Scholar profiles:

Manuel Espino:
https://scholar.google.com/citations?hl=en&user=Sx49oXIAAAAJ
Daniel Fernandez-Garcia
https://scholar.google.com/citations?user=B8hhzusAAAAJ&hl=en&oi=ao
**corresponding author** :Albert Folch:

https://scholar.google.com/citations?user=fPSqvKoAAAAJ&hl=en&oi=ao

**We thank the reviewers for their valuable comments. All suggested revisions have been completed, and the manuscript has been fully revised accordingly.**

**Sincerely,**
**Jiangyue**